# Rab5 and Rab11 maintain hematopoietic homeostasis by restricting multiple signaling pathways in *Drosophila*

**Shichao Yu, Fangzhou Luo, Li Hua Jin\***

Department of Genetics, College of Life Sciences, Northeast Forestry University, Harbin, China

**Abstract** The hematopoietic system of *Drosophila* is a powerful genetic model for studying hematopoiesis, and vesicle trafficking is important for signal transduction during various developmental processes; however, its interaction with hematopoiesis is currently largely unknown. In this article, we selected three endosome markers, Rab5, Rab7, and Rab11, that play a key role in membrane trafficking and determined whether they participate in hematopoiesis. Inhibiting *Rab5* or *Rab11* in hemocytes or the cortical zone (CZ) significantly induced cell overproliferation and lamellocyte formation in circulating hemocytes and lymph glands and disrupted blood cell progenitor maintenance. Lamellocyte formation involves the JNK, Toll, and Ras/EGFR signaling pathways. Notably, lamellocyte formation was also associated with JNK-dependent autophagy. In conclusion, we identified Rab5 and Rab11 as novel regulators of hematopoiesis, and our results advance the understanding of the mechanisms underlying the maintenance of hematopoietic homeostasis as well as the pathology of blood disorders such as leukemia.

## Introduction

*Drosophila* is a powerful genetic model for studying hematopoiesis due to the conservation between its hematopoietic system and that of mammals, including conserved regulatory factors and signaling pathways (*Yu et al., 2018a*; *Banerjee et al., 2019*). By utilizing this model, we can also improve the understanding of the molecular mechanisms underlying some blood system diseases, such as leukemia. While two main waves of hematopoiesis occur in *Drosophila* during the entire life cycle, the existence of hematopoiesis in the adult stage is controversial (*Ghosh et al., 2015*; *Sanchez Bosch et al., 2019*). The first wave occurs in the embryonic head mesoderm, where two types of hemocytes, plasmatocytes and crystal cells, are derived (*Holz et al., 2003*). Plasmatocytes are macrophage-like cells that can kill invading pathogens by phagocytosis (*Tepass et al., 1994*), while crystal cells play an important role in wound healing via the melanization response (*Lanot et al., 2001*).

The lymph gland, consisting of a pair of anterior lobes and a series of posterior lobes, is the site of the second phase of hematopoiesis that occurs during the larval stage (*Jung et al., 2005*). During metamorphosis, the lymph gland dissociates and releases hemocytes into the circulating hemolymph (*Grigorian et al., 2011*). Three distinct zones are identified within the anterior lobe: the medullary zone (MZ), where prohemocytes (blood cell progenitors) reside; a cortical zone (CZ) consisting of mature hemocytes, including plasmatocytes and crystal cells; and a posterior signaling center (PSC), which controls lymph gland homeostasis under both normal conditions and immune challenge (*Jung et al., 2005*; *Yu et al., 2018a*). The balance between the maintenance and differentiation of the MZ is a complex biological process involving a series of internal and external regulators and signaling pathways, such as Wg, Janus kinase (JAK)/STAT, insulin, and ROS (*Krzemień et al., 2007*; *Sinenko et al., 2009*; *Owusu-Ansah and Banerjee, 2009*; *Benmimoun et al., 2012*). Upon wasp infestation, the lymph gland can produce another type of hemocyte, the lamellocyte, which is much

**\*For correspondence:**
lhjin2000@hotmail.com

**Competing interests:** The authors declare that no competing interests exist.

larger than other hemocyte types and rare in healthy larvae (*Lanot et al., 2001*). Lamellocytes function mainly to encapsulate foreign objects that are too large to be phagocytosed by plasmatocytes (*Rizki and Rizki, 1992*). Previous studies have shown that the JAK/STAT, JNK, Toll, Notch, and ecdysone pathways contribute to lamellocyte formation (*Sorrentino et al., 2002*; *Zettervall et al., 2004*; *Small et al., 2014*); however, the mechanism by which lamellocyte fate is specified is incompletely understood.

Rab family proteins, members of the larger family of Ras-like GTPases, play key roles in regulating vesicle trafficking and are evolutionarily conserved in many organisms (*Zhang et al., 2007*). As small GTPases, Rab proteins cycle between GTP-bound and GDP-bound forms (*Molendijk et al., 2004*; *Pfeffer and Aivazian, 2004*). To date, 31 Rab proteins have been identified and shown to be critical in multiple biological processes (*Zhang et al., 2007*). Among these proteins, Rab5 (an early endosome marker), Rab7 (a late endosome marker), and Rab11 (a recycling endosome marker) are members of the 'core Rabs' family due to their crucial roles in vesicle transport and multiple developmental processes (*Dunst et al., 2015*). For instance, Rab7 participates in wing disc dorsal/ventral pattern formation (*Wilkin et al., 2008*), whereas Rab5 and Rab11 play key roles in cellularization and dorsal closure (*Pelissier et al., 2003*; *Sasikumar and Roy, 2009*; *Mateus et al., 2011*). However, only a few reports describe the functions of these Rabs in *Drosophila* hematopoiesis and indicate that the hemocyte-specific depletion of *Rab5* and *Rab11* leads to an increase in the circulating hemocyte count (*Jean et al., 2012*; *Del Signore et al., 2017*). Furthermore, a study in mammals showed that vacuolar protein sorting protein 33b (VPS33B) mediates hematopoiesis by regulating exosomal autocrine signaling in humans and interacts with RAB11A (*Gu et al., 2016*). No additional direct evidence has been found to indicate the relationships between the three core Rabs and hematopoiesis.

In this study, we downregulated *Rab5*, *Rab7* and *Rab11* individually in the hematopoietic system and found that inhibiting *Rab5* or *Rab11* but not *Rab7* strongly induced cell overproliferation in the *Drosophila* blood system. In addition, inhibiting *Rab5* or *Rab11* in hemocytes and the CZ resulted in the loss of MZ quiescence and aberrant lamellocyte differentiation in lymph glands and circulating hemocytes due to activation of the JNK, Ras/EGFR, and Toll signaling pathways. Moreover, we found that during lamellocyte formation, JNK and Ras acted coordinately and that Toll acted downstream of JNK in this process. Finally, we showed that JNK-induced lamellocyte production upon disruption of Rab5/Rab11 GTPase activity was autophagy-dependent. These data advance our understanding of the relationship between vesicle transport and hematopoiesis as well as the mechanism underlying lamellocyte differentiation. Furthermore, our results may provide a fundamental basis for studying vesicle transport in the pathology of leukemia.

## Results

### Inhibiting *Rab5* or *Rab11* promotes cell proliferation in the blood system

To test whether Rab5, Rab7 or Rab11 functions in *Drosophila* hematopoiesis, we first inhibited the GTPase activity of these three Rabs in the blood system by crossing flies expressing the differentiated hemocyte-specific driver *Hml>UAS-GFP* with *UAS-Rab5$^{DN}$*, *UAS-Rab7$^{DN}$*, and *UAS-Rab11$^{DN}$* flies and quantified the circulating hemocyte count. We observed a significant increase in hemocytes in *Hml>UAS-GFP>UAS-Rab5$^{DN}$* and *Hml>UAS-GFP>UAS-Rab11$^{DN}$* larvae compared with those in the corresponding controls; however, the hemocyte count in *Hml>UAS-GFP>UAS-Rab7$^{DN}$* larvae did not obviously change (*Figure 1A*). Next, we confirmed these results by using three kinds of GTPase RNAi flies (*Figure 1A*). These results were consistent with those in previous reports (*Jean et al., 2012*; *Del Signore et al., 2017*), suggesting that Rab5 and Rab11 help control the circulating hemocyte count.

Given that Rab5 and Rab11 can affect the numbers of circulating hemocytes, we sought to determine whether they control homeostasis of the lymph gland or circulating hemocytes. First, we dissected lymph glands and stained them with an anti-PH3 antibody to examine mitotic activity. Significantly more PH3-positive cells were observed in *Hml>UAS-GFP>UAS-Rab5$^{DN}$* and *Hml>UAS-GFP>UAS-Rab11$^{DN}$* lymph glands than in control (*Hml>UAS-GFP>WT* and *UAS-Rab$^{DN}$ X WT*) lymph glands (*Figure 1B,K,O*); moreover, the anterior lobes were larger in *Hml>UAS-GFP>UAS-Rab5/*

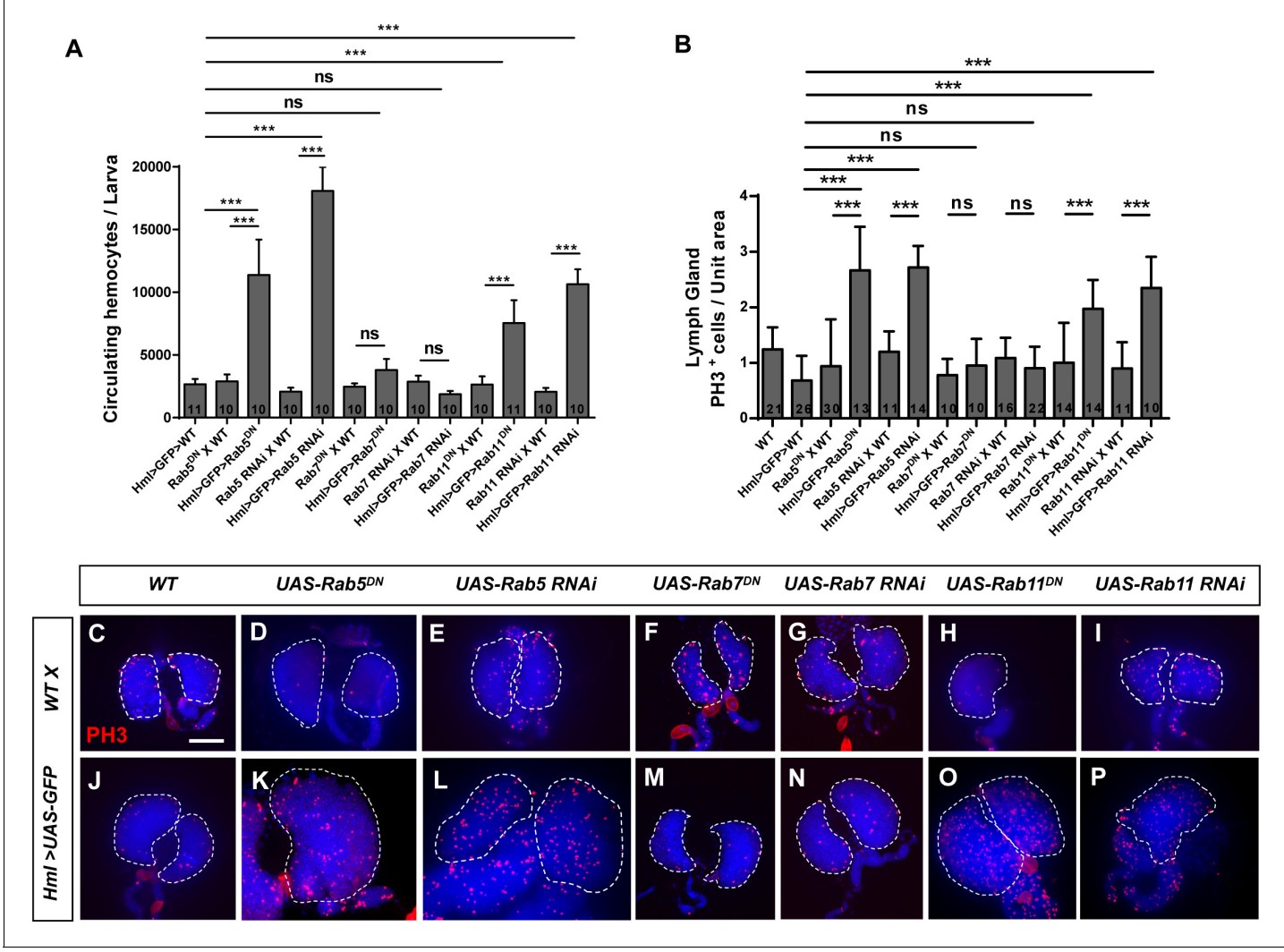

**Figure 1.** Inhibiting *Rab5* or *Rab11* promoted cell proliferation in circulating hemocytes and lymph glands. (**A**) The number of circulating hemocytes increased upon the loss of *Rab5* or *Rab11* but not *Rab7* in hemocytes. (**B–P**) Immunostaining for PH3 was performed in lymph glands from third instar larvae. The number of PH3-positive cells increased after the inactivation of *Rab5* or *Rab11* but not *Rab7*. The PH3-positive cell count per unit area was calculated as the PH3-positive cell count on a single anterior lobe divided by the area of the single anterior lobe; the quantification is shown in (**B**). Scale bar: 50 μm. ns, not significant; ***p<0.001 (one-way ANOVA).

The online version of this article includes the following figure supplement(s) for figure 1:

**Figure supplement 1.** Analyses of lymph gland size and cell proliferation in circulating hemocytes as well as the sessile hemocyte pattern.

**Figure supplement 2.** Localization of the Rab5 and Rab11 proteins in lymph glands and hemocytes.

$11^{DN}$ lymph glands than in those of the control (*Figure 1—figure supplement 1A–F,P*). Consistent with the observation that Rab7 did not control the hemocyte count, the PH3-positive cell count was comparable between *Hml>UAS-GFP>UAS-Rab7^{DN}* flies and the corresponding controls (*Figure 1B, C,F,M*). We also used RNAi lines to confirm these results (*Figure 1L,N,P*). In addition, the number of PH3-positive cells among *Hml>UAS-GFP>UAS-Rab5/11^{DN}* circulating hemocytes increased during the second instar stage but not the third instar stage (*Figure 1—figure supplement 1G–L,Q*). Given that a decreased sessile hemocyte count is associated with an increased circulating hemocyte count (*Zettervall et al., 2004*), we then assessed the sessile hemocytes from *Hml>UAS-Rab5/11^{DN}* larvae; however, the sessile hemocyte pattern was unchanged (*Figure 1—figure supplement 1M–O*), suggesting that the increased circulating hemocyte count upon inhibition of *Rab5/Rab11* resulted from the high proliferation of hemocytes. The above results indicated that both Rab5 and Rab11 maintain

hematopoietic homeostasis in *Drosophila*. Therefore, we next focused on the function of Rab5 and Rab11 in hematopoiesis.

## Rab5 and Rab11 are expressed in the *Drosophila* hematopoietic system

To examine the expression pattern of Rab5 and Rab11 in the hematopoietic system, we stained lymph glands from *Hml>UAS-GFP* (a CZ marker) larvae with anti-Rab5 and anti-Rab11 antibodies. Both Rab5 and Rab11 were expressed widely in both GFP-positive and GFP-negative areas, indicating that they were localized in both the CZ and MZ (*Figure 1—figure supplement 2A–B''*). By using *col>UAS-GFP* (a PSC marker), we observed higher levels of these two proteins in PSC cells (*Figure 1—figure supplement 2C–D''*). Moreover, the Rab5 and Rab11 proteins were localized in the cytoplasms of circulating hemocytes, while this expression pattern was not observed in *Hml>UAS-GFP>UAS-Rab5 RNAi* or *Hml>UAS-GFP>UAS-Rab11 RNAi* hemocytes (*Figure 1—figure supplement 2E–H'*). These data suggested that Rab5 and Rab11 are widely distributed throughout the *Drosophila* hematopoietic system.

## Rab5 and Rab11 in the CZ affect MZ maintenance and lamellocyte differentiation

Given that Rab5 and Rab11 affected the proliferation of lymph gland cells, we then used *Hml>UAS-GFP* to evaluate the changes in the CZ upon the downregulation of Rab5 or Rab11 GTPase activity. By determining the percentage of the GFP-positive area, we found that the CZ was larger in the *Rab5$^{DN}$* and *Rab11$^{DN}$* groups than in the control group (*Figure 2A–D*). This result was confirmed by using antibodies against the mature plasmatocyte marker P1, and expansion of P1-positive cells was observed in *Hml>UAS-Rab5/11$^{DN}$* lymph glands, including in the anterior and posterior lobes (*Figure 2E–G'*). In addition, the MZ areas were decreased in lymph glands, as detected by *dome-MESO-lacZ* (an MZ marker) (*Figure 2H–K*), indicating that Rab5 and Rab11 played a role in progenitor cell maintenance in a non-cell-autonomous manner. These results were confirmed by using *UAS-RNAi* flies (*Figure 2—figure supplement 1A–G*). Based on the current model, the non-cell-autonomous regulation of MZ maintenance is mediated by Pvr/STAT/Adgf-A signaling (*Mondal et al., 2011*). To examine whether the decreased MZ area was regulated by this signaling pathway, we overexpressed *UAS-STAT92E* and *UAS-Adgf-A* in *Hml>UAS-Rab5/11$^{DN}$* larvae and stained the lymph glands with anti-P1 antibodies. However, the increased P1-positive area was not suppressed in *Hml>UAS-Rab5/11$^{DN}$>UAS-STAT92E* or *Hml>UAS-Rab5/11$^{DN}$>UAS-Adgf-A* lymph glands (*Figure 2—figure supplement 2A–H*), suggesting that the non-cell-autonomous regulation of MZ maintenance from the CZ in our study was independent of Pvr/STAT/Adgf-A signaling; the mechanism underlying this process warrants further investigation.

Next, we examined two additional types of hemocytes, crystal cells, and lamellocytes, with anti-ProPO1 and anti-L1 antibodies, respectively. The numbers of crystal cells were comparable among the groups (*Figure 2—figure supplement 3A–C,K*); however, many lamellocytes were observed in *Hml>UAS-GFP>UAS-Rab5/11$^{DN}$* lymph glands, although lamellocytes were rare in healthy larvae (*Figure 2L–N*). This phenotype was confirmed with *UAS-RNAi* flies (*Figure 2—figure supplement 3D–F*). Accordingly, the aberrant lamellocyte differentiation was rescued after overexpression of *UAS-Rab5$^{WT}$* or *UAS-Rab11$^{CA}$* (*Figure 2—figure supplement 3G–J*). Additionally, many lamellocytes were detected in the circulating hemolymph from *Hml>UAS-GFP>UAS-Rab5/11$^{DN}$* larvae (*Figure 2O–R*), and some lamellocytes were GFP-positive (arrows in *Figure 2P,Q*), indicating that they were in a transition state between plasmatocytes and lamellocytes. Moreover, we used three additional hemocyte-specific *Gal4* drivers, *pxn-Gal4*, *He-Gal4*, and *srp-Gal4*, to confirm lamellocyte formation. Consistent with our previous result, lamellocytes emerged in lymph glands and circulating hemocytes upon the disruption of Rab5 or Rab11 GTPase activity (*Figure 2—figure supplement 4A–P*). Because the *Hml>UAS-Rab5$^{DN}$* and *Hml>UAS-Rab11$^{DN}$* phenotypes were similar, we sought to understand the relationship between Rab5 and Rab11 in the induction of lamellocyte formation. Therefore, we simultaneously inactivated *Rab5* and *Rab11* in the CZ and observed more lamellocytes in the lymph glands and circulating hemocytes, as well as larger posterior lobes (*Figure 2U*), in these groups than in the *Rab5$^{DN}$* and *Rab11$^{DN}$* groups (*Figure 2S–Y*). In addition, constitutively active Rab11 suppressed the lamellocyte formation in *Hml>UAS-Rab5$^{DN}$* lymph glands (*Figure 2—figure supplement 5A,B*). These results indicated that although Rab5 and Rab11 induce lamellocyte

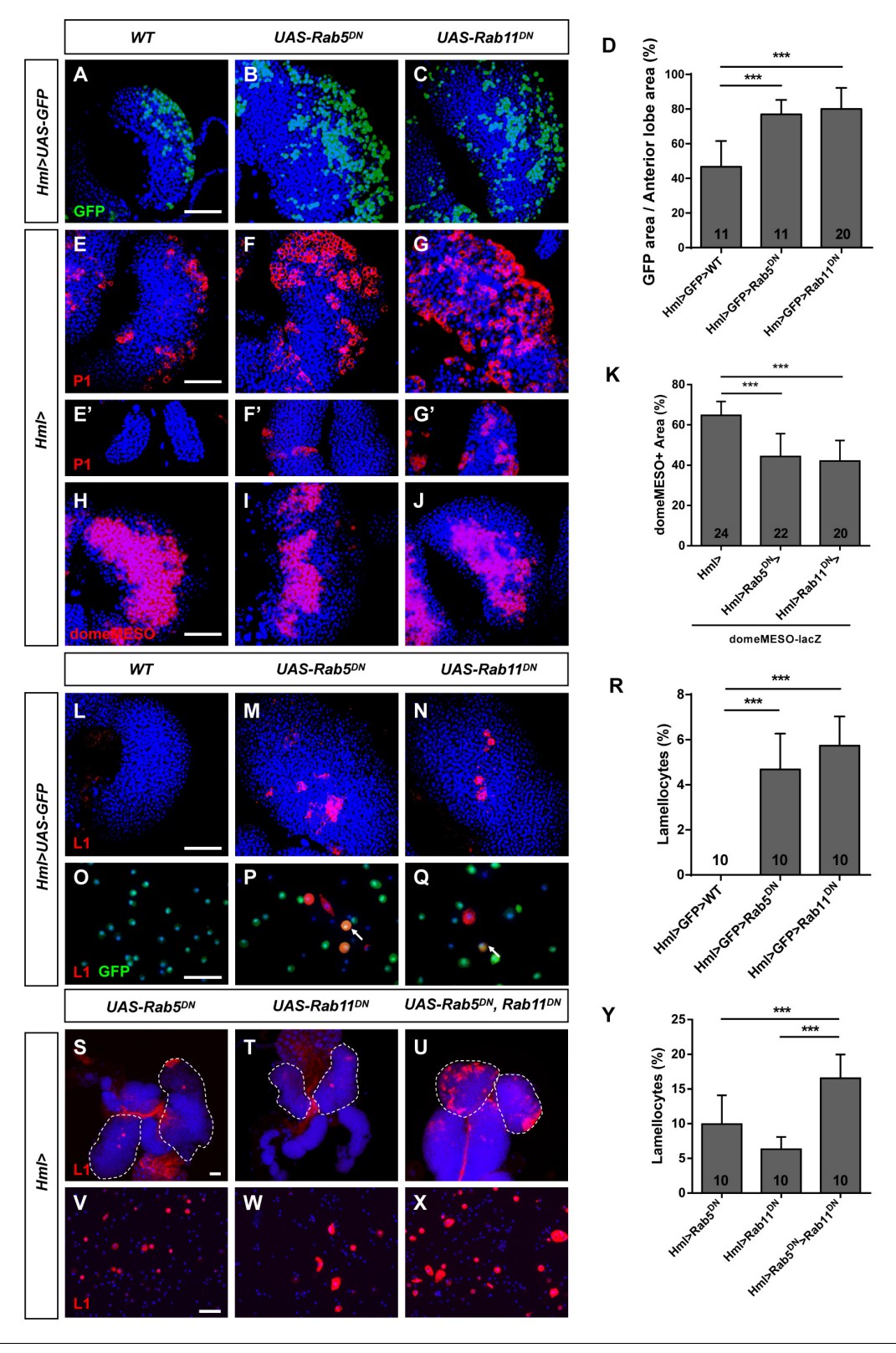

**Figure 2.** Inactivation of *Rab5* or *Rab11* promoted differentiation in circulating hemocytes and lymph glands. (A–D) The percentage of the GFP-positive area in anterior lobes was increased in *Hml>UAS-GFP>UAS-Rab5^DN* and *Hml>UAS-GFP>UAS-Rab11^DN* lymph glands; the quantification is shown in (D). (E–G′) Immunostaining for the plasmatocyte marker P1 showed that the P1-positive area was increased upon the inactivation of *Rab5* or *Rab11* in

*Figure 2 continued on next page*

*Figure 2 continued*

both the anterior (E–G) and posterior lobes (E'–G'). (H–K) Analysis using the medullary zone (MZ) marker *domeMESO-lacZ* showed that the MZ area was decreased in *Hml>UAS-Rab5$^{DN}$* and *Hml>UAS-Rab11$^{DN}$* lymph glands. The proportion of the MZ area in the anterior lobe is shown in (K). (L–Y) Immunostaining for the lamellocyte marker L1 showed that the lamellocyte count was increased in lymph glands (L–N) and circulating hemocytes (O–Q) when Rab5 or Rab11 GTPase activity was disrupted. Aberrant lamellocyte differentiation was more severe in lymph glands (S–U) and circulating hemocytes (V–X) after the simultaneous disruption of Rab5 and Rab11. The lamellocyte frequency among total circulating hemocytes is shown in (R) and (Y). Scale bar: 50 µm. ***p<0.001 (one-way ANOVA).

The online version of this article includes the following figure supplement(s) for figure 2:

**Figure supplement 1.** Evaluation of the plasmatocyte count and medullary zone (MZ) area in the lymph gland after *Rab5/Rab11* inactivation.

**Figure supplement 2.** Overexpression of *STAT92E* or *Adgf-A* did not rescue the increased P1-positive area in *Hml>UAS-Rab5/11$^{DN}$* lymph glands.

**Figure supplement 3.** Analysis of the crystal cell count and rescue assays of lamellocyte differentiation in lymph glands.

**Figure supplement 4.** Inactivation of *Rab5/Rab11* with different hemocyte-specific Gal4 drivers resulted in massive lamellocyte formation.

**Figure supplement 5.** Active Rab11 GTPase activity could restore the aberrant lamellocyte differentiation in lymph glands after inhibition of *Rab5*.

**Figure supplement 6.** Analysis of the medullary zone (MZ) and posterior signaling center (PSC) upon the inactivation of *Rab5* or *Rab11*.

formation in an independent manner, enhanced recycling endosome activity can help alleviate defects in early endosomes during this process. To determine whether the decreased MZ area was due to an altered PSC, we analyzed PSC cells with anti-Antp antibodies (*Figure 2—figure supplement 6A–C*). However, we did not find differences in the distribution or count of PSC cells after inactivation of *Rab5* and *Rab11* in the CZ relative to those in the controls (*Figure 2—figure supplement 6Y*), suggesting that the roles of these two GTPases in regulating MZ cell quiescence are PSC-independent.

Next, we used the MZ- and PSC-specific *Gal4* drivers *dome-Gal4 UAS-GFP* and *Antp-Gal4*, respectively, to inactivate *Rab5* and *Rab11* and stained lymph glands with antibodies against P1, L1, and Antp. However, the CZ area and the numbers of lamellocytes and PSC cells in the lymph gland were unchanged relative to those in the controls (*Figure 2—figure supplement 6D–U,Z,AA*). Moreover, the number of GFP-positive cells did not change when another PSC-specific *Gal4* driver, *col-Gal4 UAS-GFP*, was used (*Figure 2—figure supplement 6V–X*,BB), indicating that Rab5 and Rab11 do not play a role in regulating lymph gland hematopoiesis in the MZ or PSC. Therefore, we focused our further studies on Rab5/Rab11 function in the CZ of the lymph gland. The above results suggested that Rab5 and Rab11 in the CZ not only affect cell proliferation but also play a role in mediating the balance between MZ maintenance and differentiation.

## Inhibiting *Rab5/Rab11* can induce lamellocytes by activating the JNK pathway

Next, we sought to determine the mechanism underlying lamellocyte induction after *Rab5* or *Rab11* inactivation. JAK/STAT signaling is a crucial regulator of lamellocyte formation, and overexpression of *Drosophila JAK* in the CZ induces a large number of lamellocytes in lymph glands (*Terriente-Félix et al., 2017*). We knocked down *JAK* (known as *hop* in *Drosophila*) using *UAS-hop RNAi* to determine whether the lamellocyte count was reduced in *Hml>UAS-GFP>UAS-Rab5/11 RNAi* lymph glands. However, lamellocytes were not repressed by the knockdown of *hop* (*Figure 3—figure supplement 1A–F*), indicating that JAK signaling is not required for lamellocyte differentiation upon inhibition of Rab5 or Rab11 activity.

We then focused on the JNK pathway, which contributes to specifying lamellocyte fate (*Zettervall et al., 2004*; *Tokusumi et al., 2009*). *puckered* (*puc*) is a target gene of JNK signaling in *Drosophila*, and we found that the *puc-lacZ* signal was significantly increased in *Hml>UAS-GFP>UAS-Rab5/11 RNAi* lymph glands (*Figure 3A–C*). We also stained lymph glands with an anti-p-JNK antibody and found that the positive signal was significantly increased in the *Hml>UAS-*

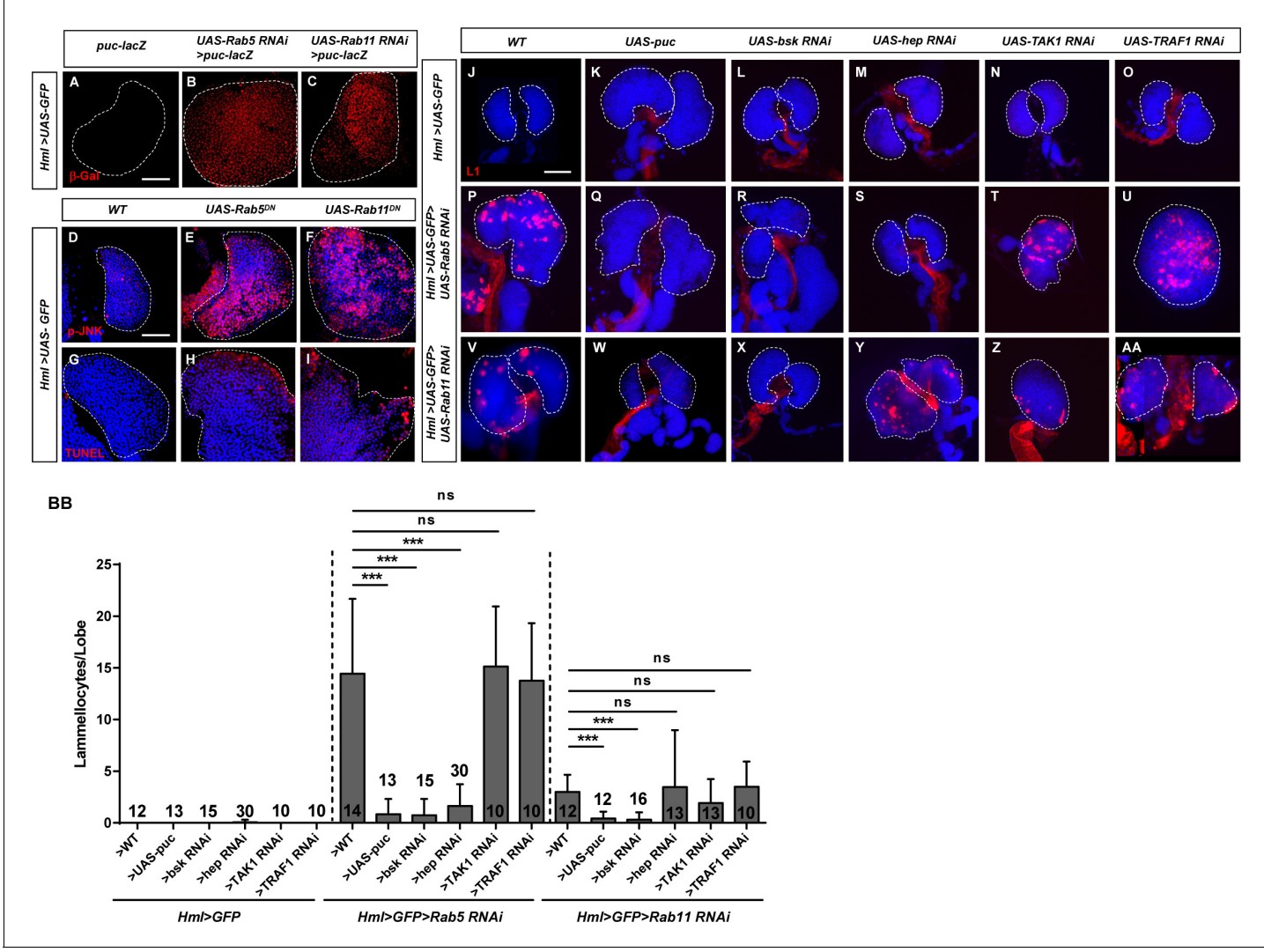

**Figure 3.** JNK signaling was activated upon *Rab5* or *Rab11* inactivation in the lymph gland. (A–I) The JNK pathway activity was elevated in lymph glands upon *Rab5* or *Rab11* inactivation, as elucidated by the monitoring of JNK signaling with *puc-lacZ* (A–C) and anti-p-JNK antibodies (D–F). Apoptotic cells in *Hml>UAS-GFP>UAS-Rab5^{DN}* and *Hml>UAS-GFP>UAS-Rab11^{DN}* lymph glands were detected by TUNEL assays (G–I). (J–AA) Immunostaining for L1 (red) showed that aberrant lamellocyte differentiation was rescued in *Hml>UAS-GFP>UAS-Rab5 RNAi>UAS-puc* (Q), *Hml>UAS-GFP>UAS-Rab11 RNAi>UAS-puc* (W), *Hml>UAS-GFP>UAS-Rab5 RNAi>UAS-bsk RNAi* (R), *Hml>UAS-GFP>UAS-Rab11 RNAi>UAS-bsk RNAi* (X), and *Hml>UAS-GFP>UAS-Rab5 RNAi>UAS-hep RNAi* (S) lymph glands. The quantifications for (J–AA) are shown in **BB**. Scale bar: 50 μm. ns, not significant; ***p<0.001 (one-way ANOVA).

The online version of this article includes the following figure supplement(s) for figure 3:

**Figure supplement 1.** Knocking down *hop* or blocking apoptosis did not repress aberrant lamellocyte differentiation.

**Figure supplement 2.** Knocking down *bsk* repressed aberrant lamellocyte differentiation in circulating hemocytes.

*GFP>UAS-Rab5/11^{DN}* groups compared with the control group (***Figure 3D–F***). Because the JNK signaling pathway is often associated with apoptosis (***Li et al., 2019***), apoptosis was assessed by transferase dUTP nick end labeling (TUNEL) assays. We did not observe a large number of apoptotic cells in *Hml>UAS-GFP>UAS-Rab5/11^{DN}* lymph glands (***Figure 3G–I***). In addition, aberrant lamellocyte differentiation was not rescued after overexpression of the antiapoptotic protein p35 (***Figure 3—figure supplement 1G–J***). To further analyze the interaction between Rab5/Rab11 and the JNK signaling pathway, the *Hml>UAS-GFP>UAS-Rab5 RNAi* and *Hml>UAS-GFP>UAS-Rab11 RNAi* lines were crossed with a series of JNK-related transgenic flies, including *UAS-puc* (the negative regulator of the JNK pathway), *UAS-bsk (JNK) RNAi*, *UAS-hep (JNKK) RNAi*, *UAS-TAK1 (JNKKK) RNAi*, and *UAS-TRAF1* (an upstream regulator of TAK1) *RNAi* flies, and the lamellocyte frequency was analyzed

(*Figure 3J–BB*). The lamellocyte frequency was significantly decreased in both *UAS-puc* and *UAS-bsk RNAi* lymph glands (*Figure 3P–R,V–X, BB*). In addition, *UAS-hep RNAi* alleviated the aberrant lamellocyte differentiation induced by the knockdown of *Rab5* but not *Rab11* (*Figure 3S,Y*), whereas *UAS-TAK1 RNAi* and *UAS-dTRAF1 RNAi* failed to reduce the lamellocyte count (*Figure 3T–U,Z–BB*). We also confirmed this result in circulating hemocytes, showing that aberrant lamellocyte differentiation was rescued in *Hml>UAS-GFP>UAS-Rab5/11 RNAi>UAS-bsk RNAi* hemocytes (*Figure 3—figure supplement 2A–E*).

To further confirm the relationship between Rab5/Rab11 and Bsk in the induction of lamellocyte formation, hemocytes were simultaneously stained with anti-Hrs (the endosome marker) and anti-p-JNK antibodies. The accumulation of endosomes and elevated p-JNK levels were observed after disrupting Rab5/Rab11 GTPase activity (*Figure 4A–C'', N–O*), and p-JNK and Hrs$^+$ endosomes exhibited more colocalization than control endosomes (*Figure 4A–C'', P*). Next, we utilized *Rab5-Gal4* and *Rab11-Gal4*, which can be used to express genes under the control of the endogenous regulatory elements of Rab5 and Rab11 loci (*Chan et al., 2011*; *Jin et al., 2012*), to overexpress *bsk* and analyzed lamellocytes in the lymph glands. We observed a large number of lamellocytes in *Rab5/Rab11>UAS-bsk* lymph glands (*Figure 4H–K*), similar to the results observed when the JNK signaling pathway was activated in mature hemocytes using *UAS-hep$^{Act}$* (*Figure 4D,E*). However, the lamellocyte count did not appreciably change upon overexpression of *bsk* in the CZ (*Figure 4F*), consistent with a previous report showing that overexpression of *bsk* in hemocytes promotes JNK phosphorylation but does not obviously affect hemocyte morphology, the sessile hemocyte population, or the number of circulating hemocytes (*Williams, 2006*). Furthermore, even after activation of *bsk* in the whole animal using a ubiquitous *Gal4* driver, no lamellocytes were found in *da>UAS-bsk* lymph glands (*Figure 4G*). In addition, hemocytes from *Rab5/Rab11>UAS-bsk*, *Hml>UAS-bsk*, and *Hml>UAS-hep$^{Act}$* larvae were stained with anti-p-JNK, anti-p-Jun (the phosphorylated form of the JNK pathway transcription factor, also known as Jra in *Drosophila*), and anti-matrix metalloproteinase 1 (Mmp1, a target gene in the JNK pathway that encodes a proteinase that cleaves proteins in the extracellular matrix) antibodies, respectively. Although the p-JNK levels were elevated in hemocytes from all experimental groups, the p-Jun and Mmp1 levels were increased in only *Hml>UAS-hep$^{Act}$* and *Rab5/11>UAS-bsk* hemocytes; no positive signal was observed in *Hml>UAS-bsk* hemocytes (*Figure 4—figure supplement 1A–U*). Furthermore, we found more Hrs and p-JNK colocalization in *Hml>UAS-hep$^{Act}$* hemocytes than in *Hml>UAS-bsk* hemocytes (*Figure 4L–M'', P*). We then sought to determine why lamellocyte formation was observable in *Rab5/11>UAS-bsk* larvae but not in *Hml>UAS-bsk* or *da>UAS-bsk* larvae. First, according to the expression data from FlyBase, *Rab5* and *Rab11* are expressed at higher levels in tissues, including fat bodies, the gut and salivary glands, than the *daughterless* (*da*) gene; we confirmed this by crossing *UAS-GFP* with *Rab5-Gal4*, *Rab11-Gal4*, and *da-Gal4* (data not shown). Therefore, we crossed *Hml>UAS-bsk* with *ppl-Gal4* (fat body-specific driver), *NP3084-gal4* (midgut-specific driver), and *elav-Gal4* (nervous system-specific driver) and stained hemocytes with anti-L1 antibodies to assess systemic effects during the process. As expected, aberrant lamellocyte differentiation was observed in *Hml>ppl>UAS-bsk* and *Hml>NP3084>UAS-bsk* circulating hemocytes but not in *Hml>elav>UAS-bsk* circulating hemocytes (*Figure 4—figure supplement 2A–D*), suggesting that systemic effects existed in *Rab5/Rab11>UAS-bsk* larvae. Consistent with the above data, the p-Jun and Mmp1 levels were increased in *Hml>UAS-Rab5/11$^{DN}$* hemocytes (*Figure 5A–F*). In addition, knocking down two transcription factors in the JNK pathway, Jra (*Drosophila* Jun) and Kay (*Drosophila* Fos), restored aberrant lamellocyte differentiation among *Hml>UAS-Rab5/11$^{DN}$* hemocytes (*Figure 5G–M*). These results indicated that Rab5 and Rab11 restrict lamellocyte production by suppressing the JNK signaling pathway and act upstream of Bsk.

## Knockdown of *Rab5/Rab11* can activate Ras/EGFR signaling to promote cell proliferation and coordinate with Bsk to regulate lamellocyte induction

Enhanced Ras/EGFR pathway activity can induce cell overproliferation (*Zettervall et al., 2004*). Given that downregulation of Rab5/Rab11 GTPase activity resulted in the high proliferation rate of lymph gland cells, we examined Ras/EGFR signaling with anti-p-Erk antibodies. Only a few positive cells were detected in control lymph glands; however, high p-Erk signaling was observed in *Hml>UAS-Rab5/11$^{DN}$* lymph glands (*Figure 6A–C*). Consistent with this finding, the expanded

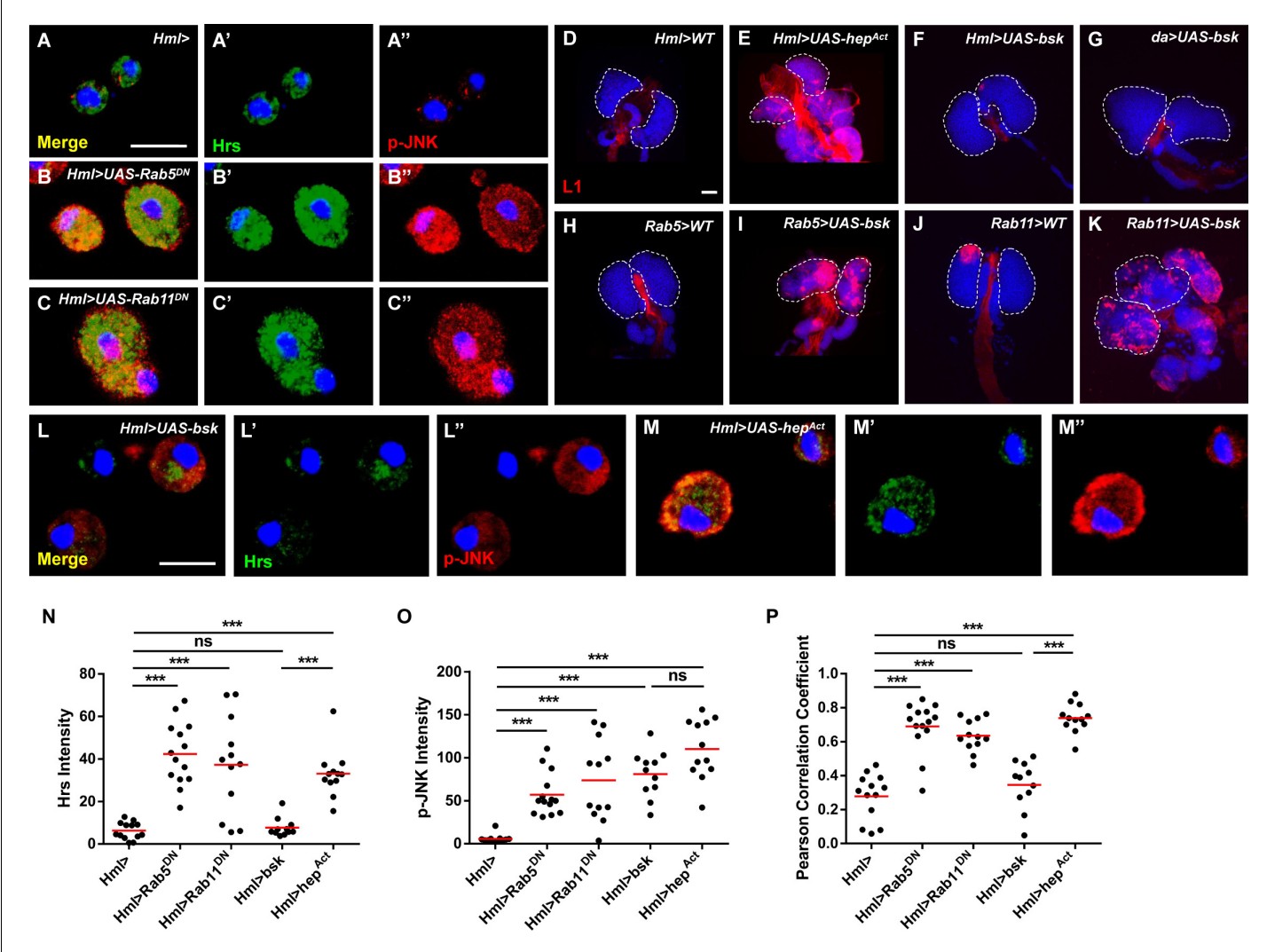

**Figure 4.** Inhibiting *Rab5* or *Rab11* induced high p-JNK levels in endosomes. (A–C'') An increased degree of Hrs (green) and p-JNK (red) colocalization was observed in *Hml>UAS-Rab5*$^{DN}$ and *Hml>UAS-Rab11*$^{DN}$ hemocytes. Merged images (Hrs+p-JNK+DAPI) are displayed in (A–C). (D–K) Massive lamellocyte formation was observed in lymph glands from *Hml>UAS-hep*$^{Act}$ (E), *Rab5>UAS-bsk* (I), and *Rab11>UAS-bsk* (K) larvae. (L–M'') *Hml>UAS-hep*$^{Act}$ hemocytes exhibited more Hrs (green) and p-JNK (red) colocalization than *Hml>UAS-bsk* hemocytes. The Hrs and p-JNK levels are shown in (N) and (O), respectively. (P) The colocalization degree between Hrs and p-JNK is displayed as the Pearson correlation coefficient, which was analyzed with the Colocalization Finder plugin from ImageJ. Scale bars: 50 µm (lymph glands) and 10 µm (hemocytes). ns, not significant; ***p<0.001 (one-way ANOVA).

The online version of this article includes the following figure supplement(s) for figure 4:

**Figure supplement 1.** Analysis of the p-JNK, p-Jun, and Mmp1 levels in circulating hemocytes.

**Figure supplement 2.** Overexpression of *bsk* in hemocytes and fat bodies or in hemocytes and the midgut simultaneously induced lamellocyte formation.

anterior lobe areas were rescued when *Ras85D* was knocked down in *Hml>UAS-Rab5/11*$^{DN}$ larvae (***Figure 6D–I***, CC). In addition, the increase in circulating hemocytes was rescued (***Figure 6DD***). Several studies have indicated that Ras/EGFR signaling functions in the PSC and MZ to induce lamellocyte formation; however, overexpressing only *Ras* in hemocytes does not significantly affect lamellocytes (***Zettervall et al., 2004***; ***Dragojlovic-Munther and Martinez-Agosto, 2013***; ***Louradour et al., 2017***). Based on our previous data showing the lack of lamellocytes when *bsk* was overexpressed in the CZ, we sought to determine whether Bsk and Ras synergistically affect lamellocyte formation. Interestingly, the lamellocyte counts were increased in *Hml>UAS-bsk>UAS-Ras*$^{V12}$

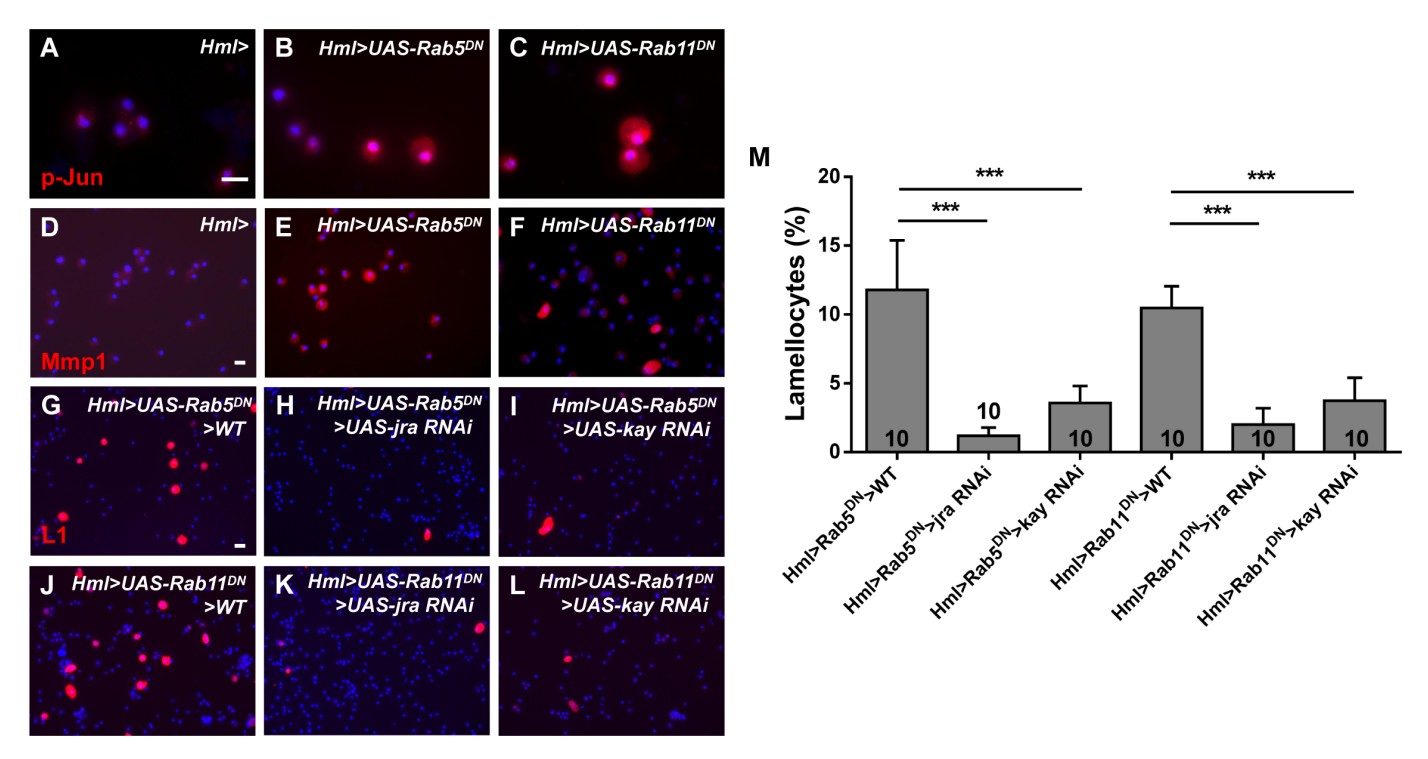

**Figure 5.** Inhibiting *Rab5* or *Rab11* in hemocytes increased the p-Jun and Mmp1 levels. (A–F) Immunostaining of circulating hemocytes showed that the levels of p-Jun (A–C) and Mmp1 (D–F) were increased after the inactivation of *Rab5* or *Rab11*. (G–M) The lamellocyte frequency, as analyzed by anti-L1 staining, was rescued in *Hml>UAS-Rab5^DN^>UAS-jra/kay RNAi* and *Hml>UAS-Rab11^DN^>UAS-jra/kay RNAi* circulating hemocytes. The percentage of L1-positive cells among total circulating hemocytes from (G–L) is shown in (M). Scale bar: 10 µm. ***p<0.001 (Student's *t*-test).

lymph glands and circulating hemocytes, while fewer L1-positive cells were detected in those of *Hml>UAS-bsk* or *Hml>UAS-Ras^V12^* (**Figure 6J–O, EE**). Moreover, knockdown of *Ras85D* in *Hml>UAS-Rab5/11^DN^* larvae partially prevented the increase in the lamellocyte count in the lymph gland (**Figure 6P–S**) and in the circulating hemocytes (**Figure 6—figure supplement 1A–E**). We also observed higher p-JNK levels in endosomes in *Hml>UAS-bsk>UAS-Ras^V12^* hemocytes than in *Hml>UAS-bsk* or *Hml>UAS-Ras^V12^* hemocytes (**Figure 6T–V'', FF**), and the p-Jun and Mmp1 levels were elevated in *Hml>UAS-bsk>UAS-Ras^V12^* hemocytes (**Figure 6W–BB**). These data suggested that Rab5/Rab11 suppresses Ras/EGFR signaling to restrict hemocyte proliferation and that Ras and Bsk coordinately promote high p-JNK levels in endosomes and eventually regulate lamellocyte induction.

## Inhibiting *Rab5/Rab11* activates the Toll pathway to induce lamellocyte formation

To assess whether the Toll pathway is involved in Rab5/Rab11-dependent lamellocyte induction, we first detected the two core components, Dif and Dorsal. We showed that Dif and Dorsal were activated in the nuclei of *Hml>UAS-Rab5/11^DN^* hemocytes, whereas nuclear localization of the two proteins was rare in the control (**Figure 7A–C,E–G**); this phenotype was also confirmed in *Hml>UAS-Rab5/11* RNAi hemocytes (**Figure 7—figure supplement 1A–F**). In addition, Dif and Dorsal were also activated in *Hml>UAS-hep^Act^* hemocytes when JNK signaling was activated (**Figure 7D,H**); this result was consistent with a previous study showing that Toll acts downstream of the JNK pathway during wing development (**Wu et al., 2015**).

Furthermore, we used *UAS-dorsal* RNAi and *UAS-Dif* RNAi to repress the Toll pathway and examined whether aberrant lamellocytes were rescued in lymph glands and hemocytes. Compared with controls, lamellocytes were decreased dramatically by knockdown of *Dif* but not *dorsal* in *Hml>UAS-Rab5/11^DN^* larvae (**Figure 7I–Y**). Based on our previous study showing that activation of Dif in lymph

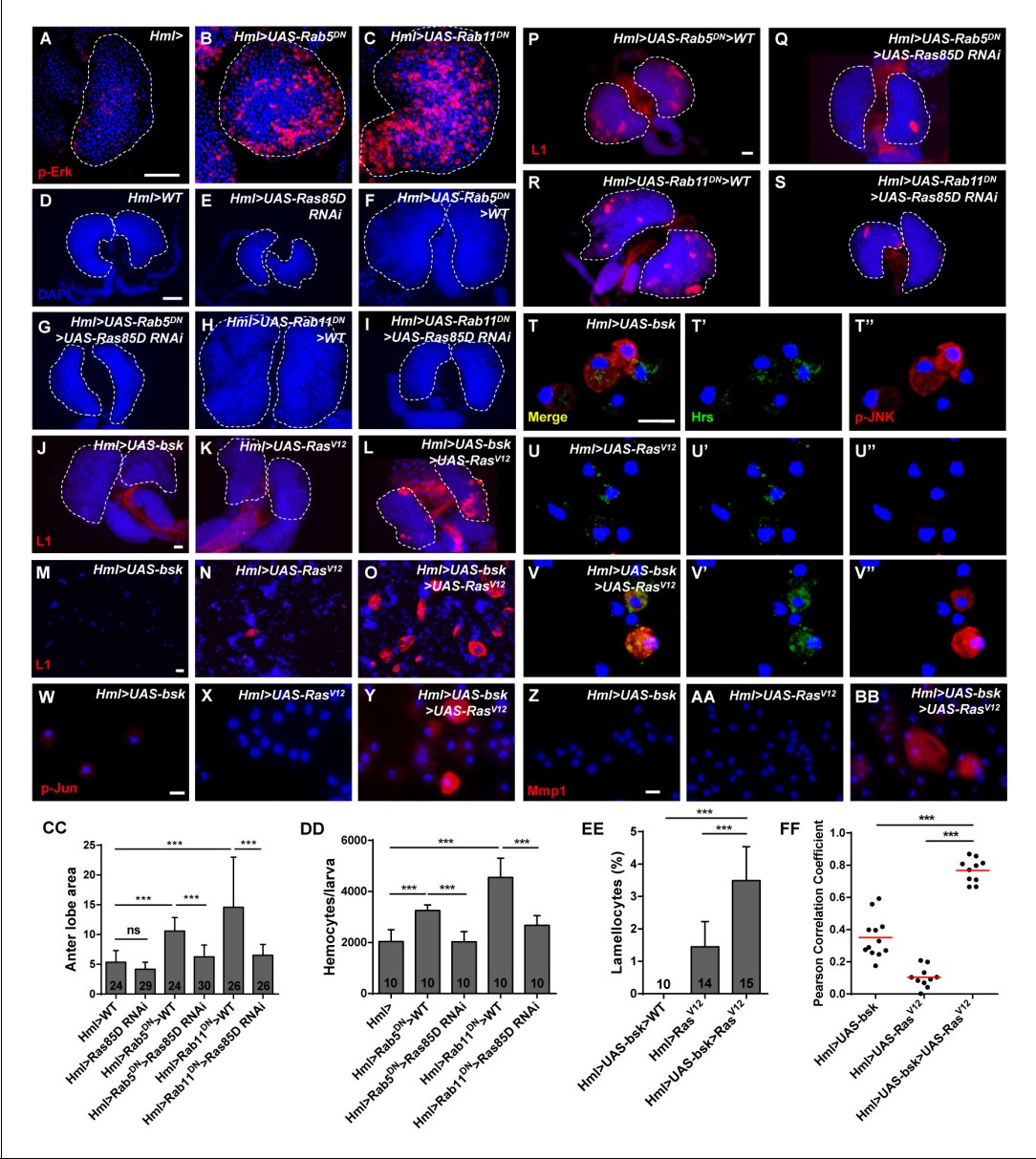

**Figure 6.** Ras/EGFR signaling was enhanced upon *Rab5* or *Rab11* inactivation. (**A–C**) Immunostaining of lymph glands showed high p-Erk (red) signals upon *Rab5* or *Rab11* inactivation. (**D–I**) Anterior lobe enlargement (visualized by DAPI staining) in *Hml>UAS-Rab5/11^DN^* lymph glands was rescued after knockdown of *Ras85D*. (**CC**) Quantification of the anterior lobe area from (**D–I**). (**J–O**) The lamellocyte count was increased in lymph glands (**J–L**) and circulating hemocytes (**M–O**) when *bsk* and *Ras* were simultaneously overexpressed in the cortical zone (CZ). (**EE**) The lamellocyte frequency in circulating hemocytes from (**M–O**). (**P–S**) The increased lamellocyte count in lymph glands after *Rab5* or *Rab11* inactivation was rescued by the knockdown of *Ras* levels using *UAS-Ras85D RNAi*. (**T–BB**) Colocalization between Hrs (green) and p-JNK (red) (**T–V''**) and p-Jun and Mmp1 (**W–BB**) was increased in *Hml>UAS-bsk>UAS-Ras^V12^* hemocytes compared with control hemocytes. The colocalization degrees are shown in (**FF**). (**DD**) Quantification of the circulating hemocyte counts in third instar larvae showed that the increased hemocyte count in *Hml>UAS-Rab5/11^DN^* larvae was rescued by the downregulation of *Ras85D*. Scale bars: 50 μm (lymph glands) and 10 μm (hemocytes). ns, not significant; ***p<0.001 (one-way ANOVA).

The online version of this article includes the following figure supplement(s) for figure 6:

**Figure supplement 1.** Knocking down *Ras85D* restored the aberrant lamellocyte differentiation in circulating hemocytes.

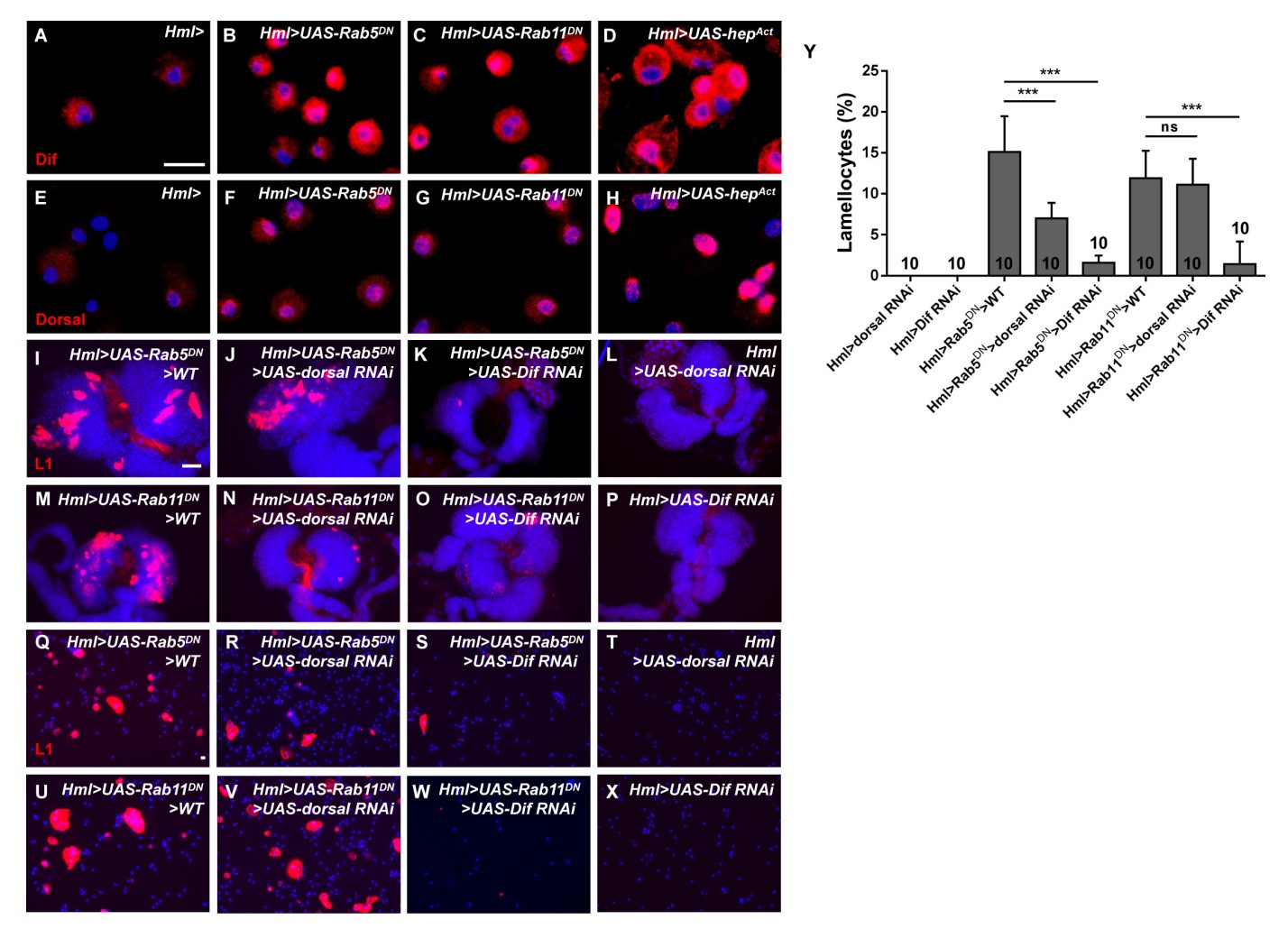

**Figure 7.** The Toll pathway was activated upon *Rab5* or *Rab11* inactivation. (A–H) Immunostaining of circulating hemocytes showed that Dif and Dorsal were activated in *Hml>UAS-Rab5/11^DN* (B–C, F–G) and *Hml>UAS-hep^Act* (D, H) hemocytes. (I–X) Detection of lamellocyte formation in hemocytes and lymph glands using anti-L1 antibodies (red) showed that *UAS-Dif RNAi* but not *UAS-dorsal RNAi* plays a significant role in restricting aberrant lamellocyte differentiation. The percentage of lamellocytes among circulating hemocytes is shown in (Y). Scale bars: 10 µm (hemocytes) and 50 µm (lymph glands). ns, not significant; ***p<0.001 (one-way ANONA).

The online version of this article includes the following figure supplement(s) for figure 7:

**Figure supplement 1.** The Toll signaling pathway was activated upon the loss of *Rab5* or *Rab11*.

glands caused lamellocyte differentiation (*Hao and Jin, 2017*), we reasoned that the disruption of Rab5/Rab11 GTPase activities induces Toll signaling pathway activation, and Dif may play a more significant role in this process.

## Autophagy participates in lamellocyte formation upon *Rab5* or *Rab11* inactivation

The JNK signaling pathway can trigger the induction of autophagy, and this process is mediated by dFOXO, the transcription factor in the JNK pathway (*Chang and Neufeld, 2010*). Overexpression of *dFOXO* in hemocytes causes lamellocyte formation in the absence of wasp parasitization (*Tokusumi et al., 2017*); we confirmed this result by using hemocytes from *pxn>UAS-GFP>UAS-dFOXO* larvae (*Figure 8—figure supplement 1A,B,S*). We also showed that dFOXO was activated in *Hml>UAS-Rab5/11^DN* larvae and that the lamellocyte count was rescued by combined *UAS-dFOXO RNAi* (*Figure 8—figure supplement 1C–J,T*); we speculated that autophagy may play a role

in this process. To further analyze the relationship between hematopoiesis and autophagy, we first detected the formation of autophagosomes in the *Cg;Atg8a-mCherry* line; generally, the presence of cytoplasmic Atg8 puncta indicates the occurrence of autophagy in a cell (*Mauvezin et al., 2014*). Under fed conditions, autophagosomes were scarce in the control group; however, large numbers of autophagosomes (visualized as red puncta in the cytoplasm) were observed in hemocytes of *Cg; Atg8a-mCherry>UAS-Rab5/11$^{DN}$* larvae (*Figure 8A–C*). These results were confirmed with the *UAS-Rab5/11 RNAi* lines (*Figure 8—figure supplement 1K–M*) and suggested that the loss of *Rab5* or *Rab11* in hemocytes can induce autophagy under fed conditions. In addition, we found that autophagosome formation tended to occur in larger hemocytes that looked like lamellocytes. To investigate this observation, we labeled autophagosomes with an anti-Atg8 antibody and labeled lamellocytes with an anti-L1 antibody. Interestingly, Atg8 puncta were observed in *Hml>UAS-Rab5/11$^{DN}$* (*Figure 8D–F*) and *Hml>UAS-hep$^{Act}$* hemocytes (*Figure 8—figure supplement 1N,O*), and Atg8-positive autophagosomes were especially prominent in lamellocytes. Ref(2)P is the fly homolog of p62 and accumulates in cells when autophagy is restricted (*Pankiv et al., 2007*; *Mauvezin et al., 2014*). Thus, we used *Cg;p62-HA* to assess autophagy upon downregulation of *Rab5* or *Rab11*. By measuring the fluorescence intensity of the HA tag, we showed that the p62 levels were decreased in *Cg;p62-HA>UAS-Rab5/11$^{DN}$* hemocytes (*Figure 8G–I,R*); the same phenotype was observed in *Cg;p62-HA>UAS-Rab5/11 RNAi* larvae (*Figure 8—figure supplement 1P–R,U*). In addition, the p62 signals were even weaker in L1-positive cells (*Figure 8H,I*). Thus, we reasoned that autophagy may participate in lamellocyte formation.

Next, we analyzed lamellocyte induction and autophagy intensity after inactivation of *Rab5* or *Rab11* at different time points (0 hr, 12 hr, 24 hr, and 48 hr) with anti-L1 antibodies and LysoTracker, respectively (*Figure 8J–Q*). Under fed conditions, lysosomes are poorly stained by LysoTracker in the fat bodies from third instar larvae, whereas induction of autophagy leads to strong punctate staining with LysoTracker (*Scott et al., 2004*). Initially, lamellocytes were not found, and the LysoTracker signals were low (*Figure 8J,N*). After 12 hr, the fluorescence intensity of LysoTracker was increased, but lamellocytes were still rare among circulating hemocytes (*Figure 8K,O*). At 24 hr, lamellocytes began to emerge, accompanied by strong LysoTracker signals in hemocytes (*Figure 8L, P*). After 48 hr, two types of lamellocytes were observed among circulating hemocytes: (1) cells with punctate L1 signals in the cytoplasm but smaller than typical lamellocytes (we defined this cell type as 'intermediates' and indicated them with asterisks in the figure) and (2) large cells with strong L1 signals (we defined this cell type as 'mature lamellocytes' and indicated them with arrows in the figure) (*Figure 8M,Q*; *Stofanko et al., 2010*). Interestingly, the LysoTracker intensity was relatively high in 'intermediates' but significantly lower in mature lamellocytes, suggesting that autophagy may promote lamellocyte formation.

To test this hypothesis, we overexpressed *Atg1* in hemocytes and analyzed lamellocyte production since enhanced Atg1 levels can directly induce autophagy and Atg1 is possibly regulated by FOXO (*Scott et al., 2007*; *Chang and Neufeld, 2010*). We used two sources of *UAS-Atg1* (*#2* and *#3*) and found large numbers of lamellocytes in both the *Hml>UAS-Atg1 #2* and *Hml>UAS-Atg1 #3* groups (*Figure 9A–C,Y*), suggesting that enhanced autophagy levels trigger the induction of lamellocyte formation. Next, we knocked down *Atg1* and *Atg8* separately in *Hml>Rab5/11$^{DN}$* larvae and evaluated the lamellocyte count. We observed fewer lamellocytes among *Hml>UAS-Rab5/11$^{DN}$>UAS-Atg1 RNAi* and *Hml>UAS-Rab5/11$^{DN}$>UAS-Atg8 RNAi* hemocytes than among control hemocytes (*Figure 9D–I,Z–AA*). To further validate that JNK is required for autophagy activation in *Hml>UAS-Rab5/11$^{DN}$* hemocytes, we repressed JNK signaling by using *UAS-bsk RNAi* or *UAS-bsk$^{DN}$* flies. The increased LysoTracker signals in *Hml>UAS-Rab5/11$^{DN}$>WT* hemocytes were suppressed in *Hml>UAS-Rab5/11$^{DN}$>UAS-bsk RNAi* and *Hml>UAS-Rab5/11$^{DN}$>UAS-bsk$^{DN}$* hemocytes (*Figure 9J–P*), suggesting that this process is JNK-dependent. Accordingly, aberrant lamellocyte differentiation was rescued in *Hml>UAS-Rab5/11$^{DN}$>UAS-bsk RNAi* and *Hml>UAS-Rab5/11$^{DN}$>UAS-bsk$^{DN}$* circulating hemocytes (*Figure 8—figure supplement 2A–G*). Next, we investigated whether lysosomal function was crucial for autophagy activation and lamellocyte formation upon the inhibition of *Rab5/ Rab11*. We then knocked down *Syntaxin 17* (*Syx17*), a SNARE that is required for the fusion of autophagosomes and lysosomes (*Takáts et al., 2013*), in *Hml>UAS-Rab5/11$^{DN}$* larvae and stained hemocytes with LysoTracker and anti-L1 antibodies. The LysoTracker intensity was decreased to a greater extent in *Hml>UAS-Rab5/11$^{DN}$>UAS-Syx17 RNAi* hemocytes (*Figure 9Q–T*), indicating that the lysosomal activity/autophagy was suppressed. In addition, the increased lamellocyte count was

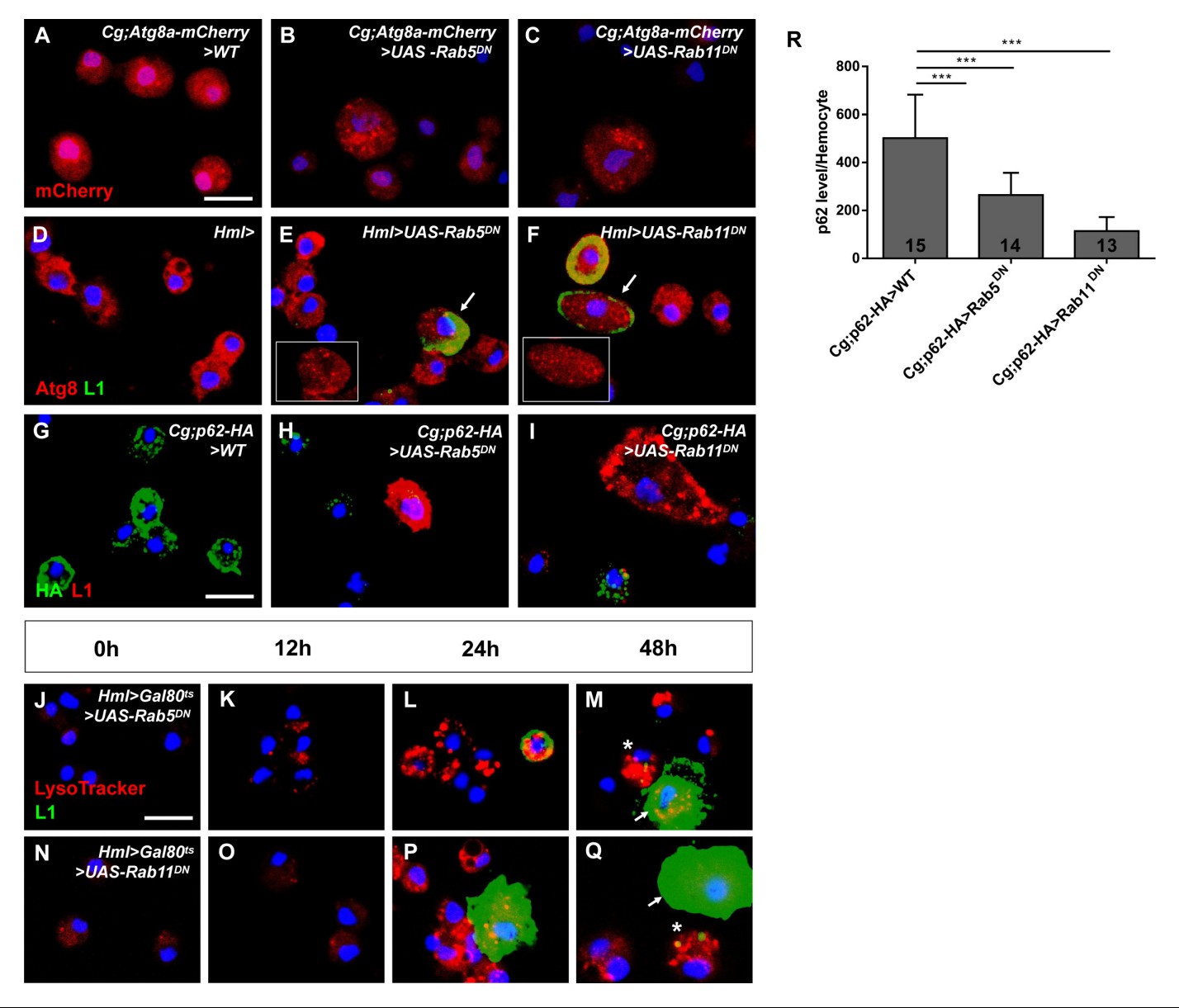

**Figure 8.** Loss of *Rab5* or *Rab11* activated autophagy in hemocytes. (**A–C**) Many autophagosomes were observed in *Cg;Atg8a-mCherry>UAS-Rab5^DN* and *Cg;Atg8a-mCherry>UAS-Rab11^DN* hemocytes. (**D–I**) Circulating hemocytes of *Hml-Gal4, Hml>UAS-Rab5^DN*, and *Hml>UAS-Rab11^DN* larvae were labeled with anti-Atg8 (red) and anti-L1 (green) antibodies, whereas those of *Cg;p62-HA>WT*, *Cg;p62-HA>UAS-Rab5^DN*, and *Cg;p62-HA>UAS-Rab11^DN* were labeled with anti-HA (green) and anti-L1 (red) antibodies (**G–I**). The arrows in (**E and F**) indicate lamellocytes with large numbers of autophagosomes. The average p62 levels per hemocyte as measured by evaluating HA fluorescence are shown in (**R**). (**J–Q**) The LysoTracker intensity (red) and lamellocyte differentiation (green) were examined in circulating hemocytes from *Hml>Gal80^ts>UAS-Rab5^DN* and *Hml>Gal80^ts>UAS-Rab11^DN* larvae after being shifted to 29°C for 0 hr, 12 hr, 24 hr, and 48 hr. In (**M**) and (**Q**), the asterisks indicate 'intermediates', and the arrows indicate 'mature lamellocytes'. Scale bar: 10 μm. ***p<0.001 (one-way ANOVA).

The online version of this article includes the following figure supplement(s) for figure 8:

**Figure supplement 1.** The dFOXO levels and autophagy activity were enhanced after the inactivation of *Rab5* or *Rab11* in hemocytes.

**Figure supplement 2.** Repressing JNK signaling restored the increased lamellocyte count in *Hml>UAS-Rab5/11^DN* larvae.

suppressed when *Syx17* was knocked down (*Figure 9U–X*, BB). This result indicated that autophagosome-lysosome fusion is critical for inducing autophagy and lamellocyte formation upon the loss of *Rab5/Rab11*. In conclusion, these data indicated that autophagy plays a role in lamellocyte formation upon the loss of *Rab5* or *Rab11*.

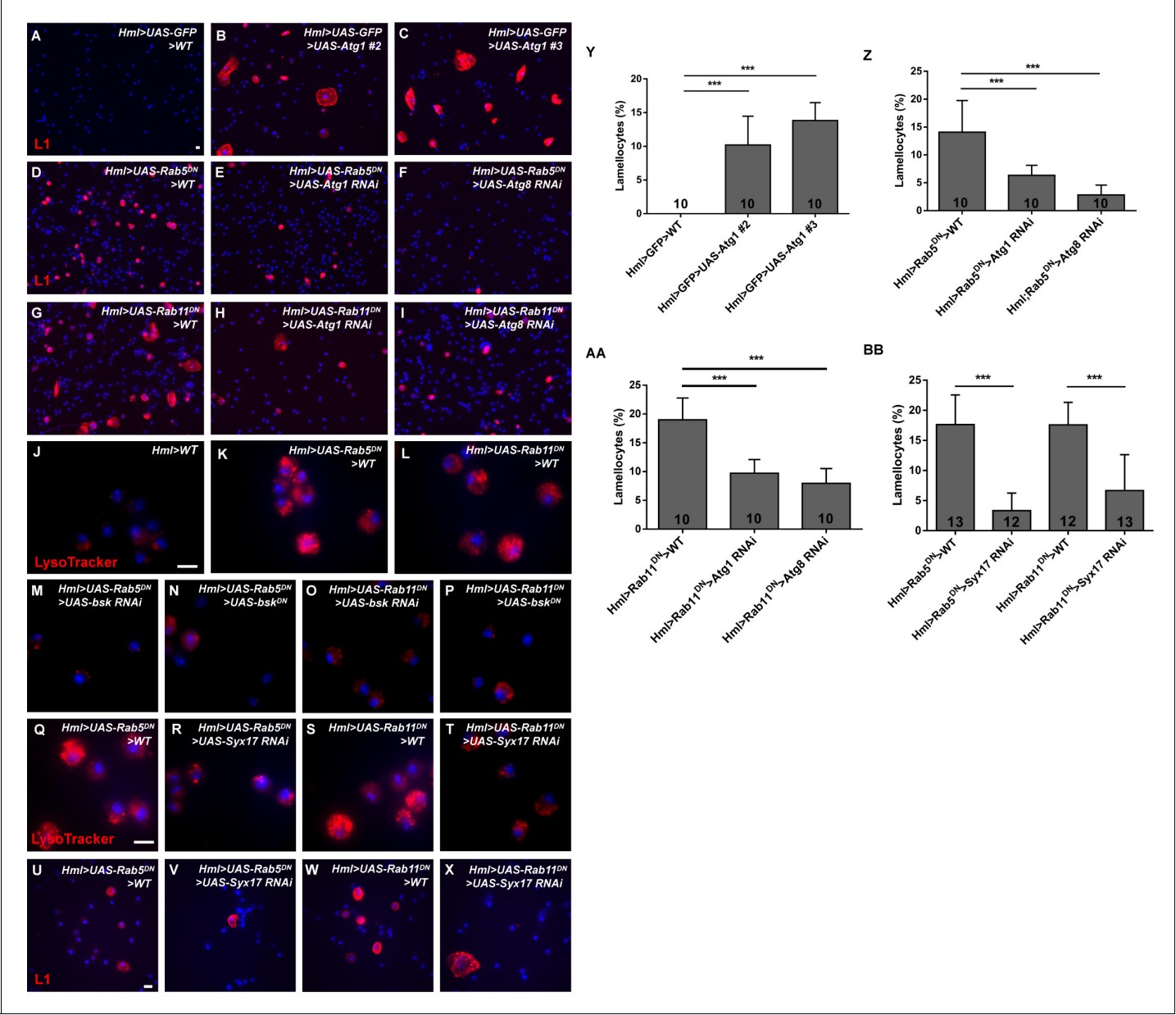

**Figure 9.** The lamellocyte formation upon the loss of *Rab5* or *Rab11* was autophagy-dependent. (A–I, U–X) The lamellocyte count was determined in circulating hemocytes by anti-L1 immunostaining. Overexpression of *Atg1* in two sources of *UAS-Atg1* flies resulted in massive lamellocyte production (A–C). In addition, knocking down *Atg1* (E, H), *Atg8* (F, I), or *Syx17* (V, X) in *Hml>UAS-Rab5/11DN* larvae suppressed the increase in the lamellocyte count. (Y–BB) The lamellocyte numbers were quantified. (J–T) LysoTracker staining in circulating hemocytes showed that the increased LysoTracker intensity was suppressed in *Hml>UAS-Rab5/Rab11DN>UAS-bsk RNAi, Hml>UAS-Rab5/Rab11DN>UAS-bskDN*, and *Hml>UAS-Rab5/Rab11DN>UAS-Syx17 RNAi* hemocytes. Scale bar: 10 µm. ***p<0.001 (one-way ANOVA).

## Discussion

Proper regulation of the hematopoietic system is crucial to the survival of both invertebrates and vertebrates. In humans, impaired hematopoietic homeostasis can cause several blood disorders, including leukemia. Multiple intrinsic and extrinsic signals contribute to the tight regulation of the blood system. In the last decade, *Drosophila* hematopoiesis has been extensively studied because it shares many conserved regulatory factors and signaling pathways with mammalian hematopoiesis (*Yu et al., 2018a*). However, the molecular mechanisms of hematopoiesis remain largely unknown. Vesicle trafficking is a critical component of signal transduction in multiple developmental processes,

and aberrant membrane transport significantly alters the signal output. Several reports have described the interaction between vesicle transport and hematopoiesis and shown that the Ras GTPase family member ARF1 modulates *Drosophila* hemocyte homeostasis via multiple signaling pathways (*Khadilkar et al., 2014*; *Khadilkar et al., 2017*; *Sinha et al., 2013*), whereas Graf regulates plasmatocyte proliferation through the GPI-enriched endocytic compartment (GEEC) endocytosis of EGFR (*Kim et al., 2017*). In addition, loss of *Vps35*, a regulator of endocytosis, leads to increased hemocyte counts and aberrant lamellocyte differentiation, which is associated with Toll and EGFR activation (*Korolchuk et al., 2007*).

In this article, we focused on three Rab proteins, Rab5, Rab7, and Rab11, that play key roles in vesicle trafficking and determined whether they function in mediating hematopoietic homeostasis. Knockdown or inactivation of *Rab5* or *Rab11* but not *Rab7* significantly induced cell overproliferation in both circulating hemocytes and lymph glands, consistent with previous reports revealing that Rab5 and Rab11 affect cell proliferation in other tissues (*Takino et al., 2014*; *Nie et al., 2019*). However, the increased lymph gland size and hemocyte count were rescued by inhibiting the Ras/EGFR pathway. We also showed that inactivation of *Rab5* or *Rab11* resulted in the loss of blood cell progenitor quiescence and aberrant differentiation of plasmatocytes and lamellocytes, which was dependent on non-cell-autonomous regulation from the CZ and independent of the PSC (*Figure 10A*). However, the crystal cell count in lymph glands was not significantly altered. Lamellocytes, a large and disc-shaped type of hemocyte, are rarely seen in healthy larvae; however, to resist wasp egg infestation, *Drosophila* will produce numerous lamellocytes that can encapsulate invading eggs, followed by the formation of a multilayer structure to kill them (*Rizki and Rizki, 1992*). PSC-derived ROS are important during this process. First, wasp parasitism elevates the ROS levels in the PSC, which can trigger the secretion of Spitz (an EGFR pathway ligand), subsequently inducing the conversion of circulating hemocytes into lamellocytes (*Sinenko et al., 2011*). In addition, a recent study showed that high ROS levels in PSC cells can also activate the Toll/NFkB pathway and eventually promote the differentiation of lamellocytes (*Louradour et al., 2017*). Moreover, starvation or a high-sugar diet results in lamellocyte differentiation in both lymph glands and circulating hemocytes (*Shim et al., 2012*; *Yu et al., 2018b*). Thus, the appearance of lamellocytes can be an indicator of a hazardous environment; however, the mechanism underlying the differentiation of lamellocytes is poorly understood.

In this article, we sought to clarify the regulatory network involved in the induction of lamellocyte formation. In *Hml>UAS-GFP>UAS-Rab5/11$^{DN}$* circulating hemocytes, a subset of L1$^+$ cells was colabeled with GFP, further confirming the plasticity of plasmatocyte conversion into lamellocytes (*Stofanko et al., 2010*); however, the cause of this conversion is not fully understood. A previous directed screen for genes involved in *Drosophila* hemocyte activation suggested that the JAK/STAT, JNK, and Toll signaling pathways participate in lamellocyte formation (*Zettervall et al., 2004*). Via rescue assays using various *UAS-RNAi* flies, we showed that both Rab5 and Rab11 restrict lamellocyte induction by repressing JNK and Toll activation rather than JAK/STAT activation. Inactivation of *Rab5* or *Rab11* significantly promoted the phosphorylation of JNK in lymph glands and hemocytes; we also observed high p-JNK levels in Hrs-positive endosomes in circulating hemocytes.

Ras/EGFR is considered a key pathway in the regulation of cell proliferation and growth; however, the role of Ras/EGFR in the CZ is unclear. We showed an increase in p-Erk activity when either *Rab5* or *Rab11* was inactivated in the CZ, and deletion of *Ras85D* in the CZ rescued aberrant lamellocyte differentiation; however, overexpression of *Ras* failed to induce massive lamellocyte differentiation. Interestingly, when *bsk* and *Ras* were overexpressed simultaneously in the blood system, many lamellocytes were produced in circulating hemocytes and lymph glands. In addition, we detected increased p-JNK levels in endosomes and increased p-Jun and Mmp1 levels in *Hml>UAS-bsk>UAS-Ras$^{V12}$* hemocytes, consistent with a previous article describing the role of Fos in regulating gut cell proliferation via the integration of EGFR and JNK signaling (*Biteau and Jasper, 2011*); however, the interaction between these two pathways warrants further study.

Toll signaling, a traditional innate immune signaling pathway, plays an important role in killing invading pathogens by triggering the production of multiple antimicrobial peptides (AMPs) (*Lemaitre and Hoffmann, 2007*). In addition, the Toll pathway is involved in lamellocyte formation. Our previous study reported that the loss of *jumu*, a member of the forkhead transcription factor family, induces lamellocyte differentiation via the nuclear translocation of Dif throughout the entire lymph gland (*Hao and Jin, 2017*). In this paper, downregulation of *Dif* rescued aberrant lamellocyte

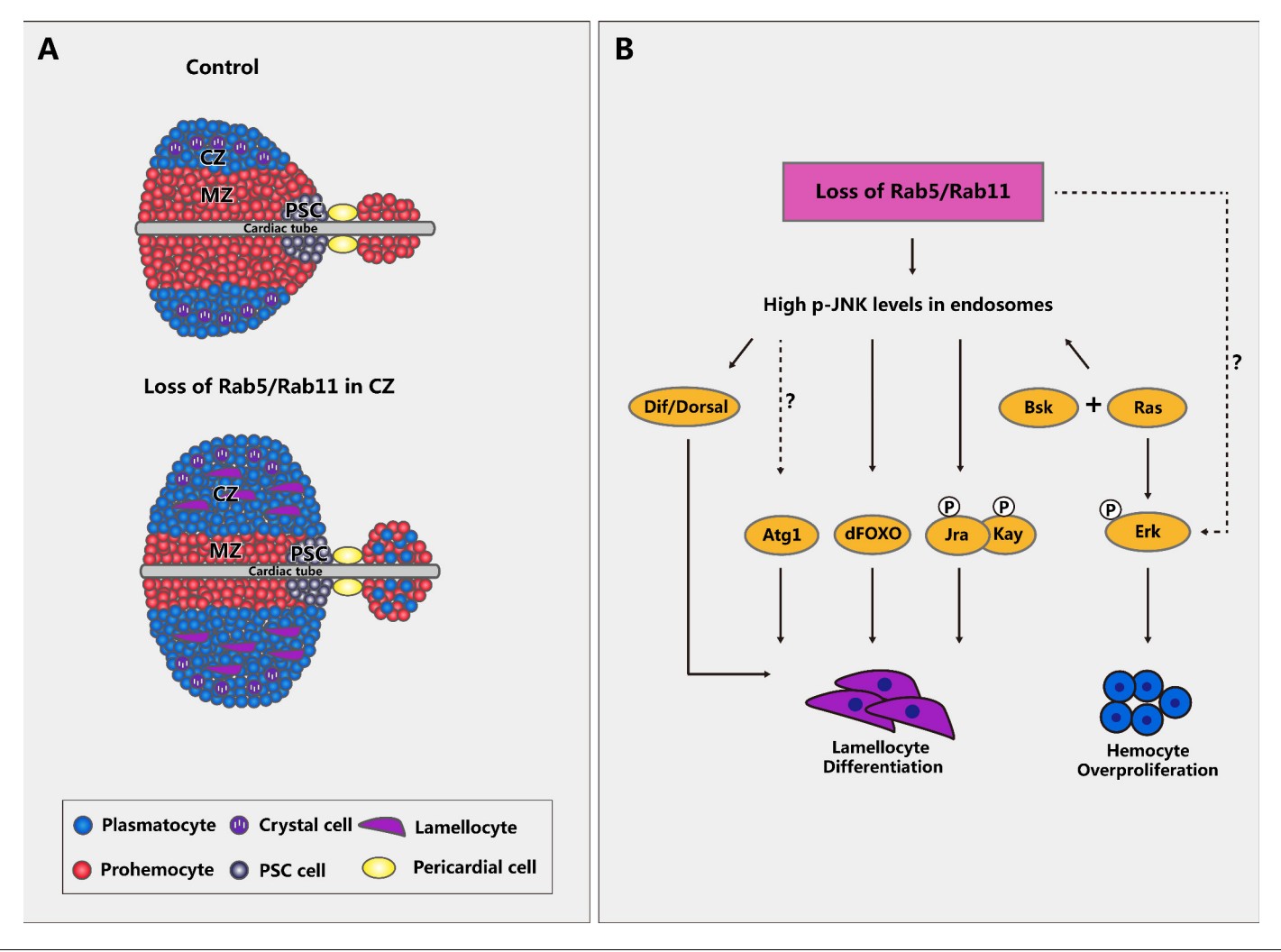

**Figure 10.** Schematic diagram of lymph gland morphology and the regulatory network of signaling pathways upon *Rab5*/*Rab11* inactivation. (**A**) Schematic diagram of lymph glands from control and *Rab5*/*Rab11*-deficient third instar larvae. When *Rab5* or *Rab11* was downregulated in the cortical zone (CZ), the formation of many premature hemocytes and lamellocytes was induced in the anterior and posterior lobes, whereas the medullary zone (MZ) area was decreased. Different cell types are represented by different colors and shapes. (**B**) Inhibiting *Rab5* or *Rab11* led to high p-JNK levels in endosomes. Next, activation of multiple signaling pathways, including JNK, Ras/EGFR, and Toll, ultimately contributed to lamellocyte differentiation and cell overproliferation.

differentiation in *Hml>UAS-Rab5/11ᴰᴺ* lymph glands, further confirming the crucial role of Toll signaling molecules, especially Dif, in lymph gland hematopoiesis. Based on a study on *Drosophila* wings showing the interaction between JNK and Toll (*Wu et al., 2015*), our findings show that Dif acts as a downstream regulator of the JNK pathway in hemocyte activation. In summary, disruption of vesicle trafficking by inactivation of *Rab5* or *Rab11* in the blood system can activate multiple signaling pathways and consequently disrupt immune cell quiescence (*Figure 10B*). Endosomal recycling is crucial for receptor/ligand control. A previous study indicated that the loss of *Rab5* leads to the stimulation of Eiger (the ligand of JNK pathway)/EGFR at the membrane of mutant cells, and Rab5 is important for eye disc development (*Takino et al., 2014*). In addition, another study showed that the activation of Rab11 GTPase in *orcl* mutants induces the stimulation of Spz, the Toll pathway ligand, and ORCL, which is required for maintaining hemocyte quiescence (*Del Signore et al., 2017*). Based on these studies, we reasoned that the phenotype observed in our study may also be due to the misregulation of various receptors, which warrants further investigation.

Autophagy is a conserved process in which cytoplasmic proteins or organelles are degraded in lysosomes. In *Drosophila*, only one report to date has revealed the function of *Atg* genes in hematopoiesis; this report showed that *Atg6* is required for not only multiple vesicle trafficking pathways but also hematopoietic homeostasis (*Shravage et al., 2013*). Here, we showed that *dFOXO* expression and autophagy were induced when *Rab5* or *Rab11* was downregulated under fed conditions. In addition, we observed increased autophagy activity in lamellocytes and found that knockdown of *dFOXO*, *Atg1*, *Atg8*, and *Syx17* restricted lamellocyte formation upon the loss of *Rab5* or *Rab11*. Interestingly, we also observed very low lysosomal activity in fully mature lamellocytes and high lysosomal activity in intermediates. We reasoned that the formation of lamellocytes requires high autophagy levels and that after lamellocyte maturation, autophagy is halted and maintained at basal levels. Moreover, we are the first to prove that overexpression of *Atg1* results in the appearance of a large number of lamellocytes. Previous studies have reported the role of Rab5 and Rab11 in autophagy in fat bodies and have shown that Rab5 promotes the degradation of autophagic cargo through lysosomal function (*Hegedűs et al., 2016*), whereas Rab11 facilitates the fusion of endosomes and autophagosomes (*Szatmári et al., 2014*). In this study, we demonstrated that deletion or inactivation of *Rab5* or *Rab11* induced autophagy under normal conditions in hemocytes and affected hematopoiesis; the underlying mechanism by which autophagy regulates hematopoiesis needs to be further clarified. In summary, we showed that maintaining a normal level of vesicle trafficking as well as autophagy in the blood system are important for hematopoietic homeostasis. We have identified Rab5 and Rab11 as novel regulators of hematopoiesis.

## Materials and methods

### *Drosophila* strains and culture conditions

UAS-Rab5$^{DN}$, UAS-Rab7$^{DN}$, UAS-Rab11$^{DN}$, UAS-Rab5$^{WT}$, UAS-Rab11$^{CA}$, puc-lacZ, UAS-hep$^{Act}$, UAS-Ras$^{V12}$, UAS-bsk, UAS-Atg1 #2, UAS-p35, Rab5-Gal4, Rab11-Gal4, UAS-GFP, ppl-Gal4, and da-Gal4 were obtained from the Bloomington *Drosophila* Stock Center (BDSC). UAS-Rab5 RNAi, UAS-Rab7 RNAi, UAS-Rab11 RNAi, UAS-bsk RNAi, UAS-hep RNAi, UAS-dorsal RNAi, UAS-Dif RNAi, UAS-Atg1 RNAi, UAS-Atg8 RNAi, and UAS-hop RNAi were obtained from the Vienna *Drosophila* Resource Center. Hml-Gal4, UAS-jra RNAi, UAS-kay RNAi, UAS-dFOXO RNAi, UAS-Ras85d RNAi, UAS-Syx17 RNAi, elav-Gal4, and Tub-Gal4; Tub-Gal80$^{ts}$ were obtained from the Tsinghua Fly Center. Hml-Gal4; UAS-2xEGFP, UAS-STAT92E, UAS-Adgf-A, and Antp-Gal4 were gifts from Utpal Banerjee (*Mandal et al., 2007*); col-Gal4 UAS-mCD8GFP was a gift from Lucas Waltzer (*Benmimoun et al., 2015*); the srp-Gal4, domeMESO-lacZ, and dome-Gal4;UAS-2xEGFP lines were gifts from Jiwon Shim; and pxn-Gal4;UAS-GFP and He-Gal4;UAS-GFP were gifts from Norbert Perrimon. UAS-puc, UAS-bsk$^{DN}$, UAS-TRAF1 RNAi, and UAS-TAK1 RNAi were gifts from José Carlos Pastor-Pareja (*Willsey et al., 2016*). Cg;Atg8a-mCherry and Cg;p62-HA were gifts from Chao Tong. The UAS-Atg1 #3 line was gifted by Guangchao Chen (*Chen et al., 2008*). The UAS-dFOXO line was a gift from Pierre Léopold. The NP3084-Gal4 was a gift from Bing Zhou (*Xiao et al., 2019*). The w$^{1118}$ line (from BDSC) was used as the wild-type (WT) line in this article. For crosses performed with the UAS/Gal4 system, flies were allowed to lay eggs at 25°C; after the eggs hatched into first instar larvae, the larvae were transferred to 29°C. The other strains were reared at 25°C. All strains and crosses were cultured on standard cornmeal-yeast medium.

### Immunostaining

For lymph gland staining, lymph glands were dissected in ice-cold PBS. After fixation for 30 min in 4% paraformaldehyde, dissected tissues were incubated in blocking solution (PBS containing 0.1% Tween-20% and 5% goat serum) for 30 min and then incubated with primary antibodies diluted in blocking solution at 4°C overnight. After several rinses in PBST, lymph glands were sequentially incubated with secondary antibodies for 2 hr and 4′6-diamidino-2-phenylindole (DAPI) for 10 min before being mounted with Slowfade mounting reagent (Thermo Fisher). For the immunostaining of hemocytes, 10 larvae were bled in 10 µl of PBS and then transferred to an adhesive glass slide. After incubation for 30 min in a humidified chamber, hemocytes were fixed using 4% paraformaldehyde for 10 min, blocked in blocking buffer, and sequentially incubated with primary antibodies, secondary antibodies, and DAPI before being mounted. All samples were observed under a Zeiss Axioskop 2 Plus

microscope or a Zeiss LSM510 confocal microscope. The following primary antibodies were used in the study: rabbit anti-PH3 (Millipore, RRID:AB_1977177), mouse anti-P1 and mouse anti-L1 (gifts from Istvan Andó); rabbit anti-ProPO1 (a gift from Erjun Ling); rabbit anti-Dif (a gift from Dominique Ferrandon); mouse anti-Antp (RRID:AB_528082), mouse anti-Dorsal (RRID:AB_528204), mouse anti-Hrs (RRID:AB_2618261) and mouse anti-Mmp1 (RRID:AB_579782) (Developmental Studies Hybridoma Bank); rabbit anti-Rab5 (RRID:AB_882240) and rabbit anti-GABARAP (anti-Atg8) (Abcam, RRID:AB_10861928); mouse anti-Rab11 (BD Biosciences, RRID:AB_397983); rabbit anti-HA (Sigma, RRID:AB_260070); rabbit anti-dFOXO (a gift from Pierre Léopold); rabbit anti-p-Erk (Cell Signaling Technology, RRID:AB_2315112); mouse anti-p-Jun (Santa Cruz, RRID:AB_629275), mouse anti-β-Gal (RRID:AB_430877), and mouse anti-p-JNK (RRID:AB_430864) (Promega). All Alexa Fluor 488- and Alexa Fluor 568-conjugated secondary antibodies were purchased from Thermo Fisher.

### LysoTracker staining

First, 5–10 larvae were bled in 10 µl of PBS, and circulating hemocytes were then incubated with 1 µM LysoTracker Red DND-99 (Thermo Fisher) for 30 min and with 4% paraformaldehyde for 10 min. Finally, the hemocytes were mounted with Slowfade mounting reagent and analyzed using a Zeiss Axioskop 2 Plus microscope or a Zeiss LSM 510 Meta confocal microscope. Flies carrying the Gal80$^{ts}$ element were first reared at 18°C and then shifted to 29°C for 0 hr, 12 hr, 24 hr, and 48 hr to activate the *Gal4* driver.

### Quantification of circulating hemocytes

Groups of five wandering third instar larvae (at least 10 groups per treatment) were bled in 20 µl of PBS and transferred to a Neubauer Improved hemocytometer (Marienfeld) to quantify the number of circulating hemocytes per larva.

### TUNEL assays

TUNEL assays (Roche) were employed according to the manufacturer's instructions to detect apoptosis.

### Statistical analysis

The PH3$^+$, ProPO$^+$, L1$^+$, or Antp$^+$ cell counts were quantified with ImageJ. For the quantification of the area of GFP$^+$ or domeMESO$^+$ cells, the images were converted to eight bits and adjusted to obtain an identical threshold by using ImageJ. Then, the area with the identical threshold was measured as the fluorescence$^+$ area. The fluorescence intensity of p62-HA was also analyzed with ImageJ. The p-values for all experiments were calculated with a two-tailed, unpaired Student's *t*-test or one-way ANOVA using GraphPad Prism 6.0 software. The thresholds for statistical significance were established as $*p<0.05$, $**p<0.01$, and $***p<0.001$. The error bars in all column diagrams indicate the means ± SDs. All staining experiments were performed at least three times independently. For each genotype in each independent experiment, at least 10 lymph glands or at least 10 images of hemocytes were analyzed. The detailed sample sizes are shown either directly on the bar or by dot plots.

## Acknowledgements

We thank Utpal Banerjee, Jiwon Shim, Lei Xue, Lucas Waltzer, Norbert Perrimon, José Carlos Pastor-Pareja, Chao Tong, Bing Zhou, and Guangchao Chen for gifting the fly strains and Istvan Andó, Dominique Ferrandon, Erjun Ling, Pierre Léopold, and the Developmental Studies Hybridoma Bank for providing the antibodies. We thank the Bloomington *Drosophila* Stock Center, Vienna *Drosophila* RNAi Center, and TsingHua Fly Center for providing the fly stocks. We also thank Neal Silverman for the valuable suggestions on this paper. This work was supported by the National Natural Science Foundation of China (31772521).

## Additional information

### Funding

| Funder | Grant reference number | Author |
| --- | --- | --- |
| National Natural Science Foundation of China | 31772521 | Li Hua Jin |

The funders had no role in study design, data collection and interpretation, or the decision to submit the work for publication.

### Author contributions

Shichao Yu, Investigation, Visualization, Writing - original draft; Fangzhou Luo, Investigation, Visualization; Li Hua Jin, Supervision, Funding acquisition, Project administration, Writing - review and editing

### Author ORCIDs

Li Hua Jin https://orcid.org/0000-0001-5912-9800

### Decision letter and Author response

Decision letter https://doi.org/10.7554/eLife.60870.sa1
Author response https://doi.org/10.7554/eLife.60870.sa2

## Additional files

### Supplementary files

• Transparent reporting form

### Data availability

All data generated or analysed during this study are included in the manuscript and supporting files.

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
