## [Decision Letter]

**Acceptance summary:**

This study dissects the role of Rab proteins in *Drosophila* hematopoietic homeostasis and shows that Rab5 and Rab11 are implicated in the hemocyte proliferation and differentiation through several genetic pathways, including the *JNK*, Toll, and Ras/EGFR. Moreover, inhibition of Rab/Rab11 alters autophagy in hemocytes, together activating differentiation of lamellocytes. Genetic interactions indicated in this study will promote further investigations of intricate balances in the endocytic machinery from the perspective of immunity and hematopoiesis.

**Decision letter after peer review:**

Thank you for submitting your article "Rab5 and Rab11 maintain hematopoietic homeostasis by restricting multiple signaling pathways in *Drosophila*" for consideration by *eLife*. Your article has been reviewed by three peer reviewers, one of whom is a member of our Board of Reviewing Editors, and the evaluation has been overseen by Utpal Banerjee as the Senior Editor. The following individual involved in review of your submission has agreed to reveal their identity: Lucas Waltzer (Reviewer #2).

The reviewers have discussed the reviews with one another and the Reviewing Editor has drafted this decision to help you prepare a revised submission.

Summary:

In this manuscript, Shichao and Jin investigated the hematopoietic phenotype driven by the loss of endosomal regulators, Rab5 and Rab11. Inhibition of Rab5 or Rab11 induces hyperproliferation of the lymph gland or circulating hemocytes and activates the lamellocyte differentiation. Multiple signaling pathways including the *JNK*, Ras/EGFR, Toll, and autophagy are altered by the loss of endocytic control, highlighting the importance of vesicle trafficking in the *Drosophila* hematopoiesis.

All the reviewers find that the study is well-executed, and they were quite enthusiastic about the manuscript. But, they also find that there are several conclusions to be significantly strengthened to clearly support the authors' claim. In addition to 5 essential revisions, it is recommended that the authors look for bringing new insights into the Rab5/11 mechanism.

Essential revisions:

1) Rab5/Rab11 and autophagy

A general concern is at the end of the study where autophagy is interrogated. Unlike the previous sections of the manuscript, this section appears more preliminary. A knowledge gap remains: what is the functional role of autophagy in hematopoiesis? Does the autophagy induction from Rab5/11 perturbation require functional lysosomes? If lysosome function is perturbed, does this suppress the increased autophagy phenotype? An alternative model posits that autophagy merely sequesters signaling molecules away from recycling endosome-cell surface cycling. If so, fusion of autophagsomes with lysosomes may not be necessary for the role of autophagy in lamellocyte formation. Can autophagosomal SNARES be knocked down or perturbed to see the impact on formation? Evidence is presented that Rab5/11 knockdown increases autophagy, and this is correlated with lamellocyte formation/*JNK* activation. This conclusion could be tuned-down or directly tested (using for instance *Hep* or Bsk RNAi).

2) An increase in the number of circulating hemocytes

The origin of the increased number of circulating hemocytes should be clarified. The authors do not observe an increase in proliferation in circulating hemocytes at the L3 stage (in contrast with the situation in the lymph gland). Increased proliferation may have occurred earlier. Alternatively, it is possible that the increase is merely due to a disruption of the sessile hemocyte compartment. These 2 possibilities could be easily assessed.

3) *Rab5/Rab11-gal4* expression

Different phenotypes driven by *Rab5/Rab11-gal4* require additional experiments to address why it leads to the discrepancy when compare to hemocyte-specific drivers. This could simply be due to different levels of overexpression (and a threshold effect to observe lamellocyte induction in response to *JNK* activation). Or it is possible that Rab5/11-gal4 is expressed in other tissues causing systemic effects. This needs to be assessed either by looking directly at Bsk accumulation or with a UAS-destablizedGFP. At present, the conclusion that bsk needs to be expressed "under endosomal control" to induce lamellocyte seems weak and the concept still ill-defined.

4) Mixture of lymph gland- and circulating hemocyte phenotypes

Lymph gland- and circulating hemocytes are intermingled, and some of the pathways, including the Toll pathway and autophagy, are validated only by circulating hemocytes. The mixture of two cell types throughout the manuscript makes it hard to follow the authors' rational. Moreover, hemocyte differentiation/proliferation mechanisms in two tissues may differ. The authors need to better align the two populations with more caution and confirm all the pathways in both cell types.

5) The non-cell-autonomous phenotype in the lymph gland

In Figure 2H-K, the authors showed that the expression of RNAi against Rab5 or Rab11 in the CZ decreases the MZ in a non-cell-autonomous manner. How do the authors explain this phenotype by the current model? This should be shown with additional experiments.

Physiological- or cell biological relevance of Rab5/11-mediated function

What is the physiological or cell biological relevance of the Rab5/Rab11 regulation shown in this study? It will be important to show whether the endocytic control indeed is critical for the immune response/hemocyte development to manage all the associated signals.

[Editors' note: further revisions were suggested prior to acceptance, as described below.]

Thank you for resubmitting your work entitled "Rab5 and Rab11 maintain hematopoietic homeostasis by restricting multiple signaling pathways in *Drosophila*" for further consideration by *eLife*. Your revised article has been evaluated by Utpal Banerjee (Senior Editor) and a Reviewing Editor.

The manuscript has been improved but there are some remaining issues that need to be addressed before acceptance, as outlined below:

Reviewer #1:

Yu and colleagues supplemented a large number of data and addressed most of the concerns raised by the reviewers. New data clearly resolved concerns raised and explained how circulating hemocytes are increased upon loss of Rab5/11 and why Rab5/11-gal4 differs from Hml-gal4. Despite the clarification, the biological relevance of Rab5/Rab11 signaling in hematopoiesis is still unclear. It is well-known that Rab5/Rab11 endosomal pathway controls multiple proteins involved in cellular signalings, including EGFR/Ras, Eiger/*JNK*, and Toll, and thus, in my opinion, it will be important to understanding why Rab5/Rab11-mediated control is significant in the blood.

1) Biological relevance

Although the authors extensively investigated the functions of Rab5/Rab11 in the blood cell signaling, it is still ambiguous how the endosomal control is maintained during development or immunity and why it is significant in hematopoiesis. For example, does Rab5/Rab11 control the proliferation of hemocytes in a specific developmental stage by repressing or activating Ras/Erk? Is Rab5/Rab11 inhibited by immune challenges to directly stimulate the lamellocyte differentiation? In which context Rab5/Rab11 gets modified and manipulates target signals similar to phenotypes derived by Rab5^DN^/Rab11^DN^ or Rab5/Rab11 overexpression?

2) Schematic diagram

A schematic diagram shown in Figure 10B does not fully represent the genetic pathways identified in the results. For example, there is no evidence showing a direct activation of pERK by Rab5/Rab11. And even though Rasv12 alone could not activate the lamellocyte differentiation, active Ras and Bsk together can induce the lamellocyte. The authors introduced dFOXO as a target of *JNK* and showed that loss of dFOXO in *Hml>Rab5/11^DN^* rescues the lamellocyte phenotype. These points are missing. Or the diagram could be simplified to help understand the major points.

Reviewer #2:

The authors demonstrate that decreasing the activity of Rab5 or Rab11 in differentiated hemocytes leads to an increase in blood cell number and to the spurious differentiation of lamellocytes. They show that the *JNK*, Toll and ras/EGFR pathways are implicated in lamellocyte induction downstream of Rab5/11 and further investigate *JNK*-induced autophagy in this process. These results bring important new insights into the fine tuning of blood cell homeostasis by the endocytic machinery.

In my opinion, the revisions performed by the authors clearly address the major points that were raised during the review process. The new experiments that have been added bring convincing results, even when they are "negative" (as for the non-autonomous phenotype in the lymph gland). Notably, they highlight the functional importance of autophagy/lysosomes and clarify the origin of the increase in circulating blood cells. The authors have also appropriately taken into considerations most other comments. In sum, the revised version of the manuscript is clearly improved.

Reviewer #3 :

This study dissects the role of Rab proteins Rab5 and 11 on *Drosophila* hematopoietic homeostasis. Using RNAi and dominant negative tissue-specific genetic manipulation, they demonstrate that Rab5/11 influence *Drosophila* hemocyte proliferaction in a *JNK*-dependent, autophagy dependent manner. Strengths include thorough genetic dissection of several signaling pathways in both the hemocytes and associated lymph tissues.

Comments:

1) This study primarily dissects the roles of Rab5 and Rab11 proteins in hemocyte homeostasis. After the initial review, a major concern was operationally dissecting the functional role for autophagy in hematopoiesis. The new Syx17 RNAi data demonstrates a requirement for functional autophagosome-lysosome fusion for the elevated hemocyte levels.

2) For the increased hemocyte concerns, the anti-PH3 staining indicates increases mitotic proliferation, and appears conclusive.

3) The driver differences data indicate variability in expression between the Rab5/Rab11 and da drivers. The text has been amended in accordance with this new data.

4) Regarding the differences in analysis between lymph and circulating hemocytes: lymph glands do not appear to manifest strong autophagy after starvation. These tissue distinctions appear to now be more discussed in the revision.

5) non-cell-autonomous phenotype in lymph gland: the PVR/STAT92E/adenosine signaling pathway is now tested, and found to be independent of it.

Overall conclusion: in general the revision has addressed the majority of points. The new experiments that have been added, particularly the Syx17 work and PH3 staining, are conclusive if somewhat minimalistic. The text is also adjusted to tone-down or re-interpret sections that were issues before. Point 5 remains unaddressed, as the lymph gland non-cell-autonomous phenotype was independent of PVR/STAT93E/adenosine pathway. However, I feel fully addressing this in full is outside the scope of this revision.

---

## [Author Response]

Essential revisions:1) Rab5/Rab11 and autophagyA general concern is at the end of the study where autophagy is interrogated. Unlike the previous sections of the manuscript, this section appears more preliminary. A knowledge gap remains: what is the functional role of autophagy in hematopoiesis? Does the autophagy induction from Rab5/11 perturbation require functional lysosomes? If lysosome function is perturbed, does this suppress the increased autophagy phenotype? An alternative model posits that autophagy merely sequesters signaling molecules away from recycling endosome-cell surface cycling. If so, fusion of autophagsomes with lysosomes may not be necessary for the role of autophagy in lamellocyte formation. Can autophagosomal SNARES be knocked down or perturbed to see the impact on formation? Evidence is presented that Rab5/11 knockdown increases autophagy, and this is correlated with lamellocyte formation/JNK activation. This conclusion could be tuned-down or directly tested (using for instance Hep or Bsk RNAi).

We thank the reviewers for the suggestions to further clarify the role of autophagy in lamellocyte formation. Syntaxin 17 (Syx17) is an autophagosomal SNARE that is important for the fusion of lysosomes and autophagosomes (Takáts et al., 2013). Inhibiting *Syx17* disrupts the fusion of lysosomes and autophagosomes and subsequently blocks autolysosomal degradation. To examine whether autophagy induction from Rab5/Rab11 perturbation requires functional lysosomes, we knocked down *Syx17* in *Hml>UAS-Rab5/11^DN^* larvae. We found that increased LysoTracker signals were decreased in *Hml>UAS-Rab5/11^DN^>Syx 17 RNAi* circulating hemocytes (Figure 9Q-T), showing that the increased lysosomal activity was blocked. In addition, the lamellocyte count was decreased in *Hml>UAS-Rab5/11^DN^>Syx 17 RNAi* circulating hemocytes (Figure 9U-X, BB). The above data indicated that lysosomal activity is critical to the role of autophagy in lamellocyte formation upon the inhibition of *Rab5*/*Rab11*.

To further validate that *JNK* is required for autophagy activation in *Hml>UAS-Rab5/11^DN^* hemocytes, we repressed *JNK* signaling by using *UAS-bsk RNAi* or *UAS-bsk^DN^* flies. Accordingly, the increased LysoTracker signals were restored in *Hml>UAS-Rab5/11^DN^>UAS-bsk RNAi and Hml>UAS-Rab5/11^DN^>UAS-bsk^DN^*hemocytes (Figure 9J-P), suggesting that this process is *JNK*-dependent.

Regarding the functional role of autophagy in hematopoiesis, 20 autophagy-related genes have been identified to date (https://flybase.org/reports/FBgg0000076), but only *Atg6* has been shown to affect hematopoiesis (Shravage et al., 2013). Autophagy is associated with many biological processes, including the immune response, aging and neurodegeneration. We are also interested in the detailed role of autophagy in hematopoiesis and will further focus on the mechanisms underlying how autophagy regulates hematopoiesis.

2) An increase in the number of circulating hemocytesThe origin of the increased number of circulating hemocytes should be clarified. The authors do not observe an increase in proliferation in circulating hemocytes at the L3 stage (in contrast with the situation in the lymph gland). Increased proliferation may have occurred earlier. Alternatively, it is possible that the increase is merely due to a disruption of the sessile hemocyte compartment. These 2 possibilities could be easily assessed.

To examine whether increased proliferation occurred at an earlier stage, we stained hemocytes from *Hml>UAS-GFP>UAS-Rab5/11 RNAi* 2^nd^ and 3^rd^ instar larvae with anti-PH3 antibodies. When compared with the *Hml>UAS-GFP>WT* hemocytes, the PH3-positive hemocyte count was increased in *Hml>UAS-GFP>UAS-Rab5/11 RNAi* hemocytes at the L2 stage (Figure 1—figure supplement 1G-I, Q), whereas we did not observe significantly increased proliferation at the L3 stage (Figure 1—figure supplement 1J-L, Q). In addition, we also examined sessile hemocytes from *Hml>GFP>UAS-Rab5/11^DN^* larvae, but the sessile hemocyte count was not decreased compared with that in *Hml>UAS-GFP>WT* larvae (Figure 1—figure supplement 1M-O). These data indicated that increased circulating hemocytes upon the inhibition of *Rab5*/*Rab11* resulted from the high proliferation of hemocytes and not from the decreased sessile hemocyte count.

3) Rab5/Rab11-gal4 expressionDifferent phenotypes driven by Rab5/Rab11-gal4 require additional experiments to address why it leads to the discrepancy when compare to hemocyte-specific drivers. This could simply be due to different levels of overexpression (and a threshold effect to observe lamellocyte induction in response to JNK activation). Or it is possible that Rab5/11-gal4 is expressed in other tissues causing systemic effects. This needs to be assessed either by looking directly at Bsk accumulation or with a UAS-destablizedGFP. At present, the conclusion that bsk needs to be expressed "under endosomal control" to induce lamellocyte seems weak and the concept still ill-defined.

Given that we observed lamellocytes in *Rab5/Rab11>UAS-bsk* lymph glands but not in *da>UAS-bsk* lymph glands, we crossed *da-Gal4* and *Rab5/Rab11-Gal4* with *UAS-GFP* to determine where there was a difference in expression in the whole larvae. As shown in the figure, the overall fluorescence intensity in *Rab5/Rab11-Gal4* was comparable with that in *da>UAS-GFP*, however; the fluorescence intensity was stronger in salivary glands from *Rab5/Rab11-Gal4* larvae (Author response image 1). In addition, based on gene expression data from FlyBase, *Rab5* or *Rab11* is highly expressed in the gut, fat bodies and salivary glands (*Rab5*: https://flybase.org/reports/FBgn0014010; *Rab11*: https://flybase.org/reports/FBgn0015790), while the *daughterless* gene has a mild expression pattern in these tissues (*da*: https://flybase.org/reports/FBgn0267821). These data suggest that *Rab5-Gal4* and *Rab11-Gal4* are stronger drivers in some tissues.

Moreover, to examine whether systemic effects exist, we crossed *Hml>UAS-bsk* with *NP3084-Gal4* (midgut-specific driver), *ppl-Gal4* (fat body-specific driver) and *elav-Gal4* (nervous system-specific driver) and stained hemocytes with anti-L1. We observed lamellocyte formation in *Hml>NP3084>UAS-bsk* and *Hml>ppl>UAS-bsk* hemocytes (Figure 4—figure supplement 2A-D), indicating that the interaction between hemocytes and fat bodies or midguts may play a role in aberrant lamellocyte differentiation. The above results showed that the different phenotypes in *Hml>UAS-bsk* and *Rab5/Rab11>UAS-bsk* may result from systemic effects, so our conclusion that *bsk* expression under Rab5/Rab11 control is crucial for lamellocyte formation is not correct. We have deleted the previous conclusion “These data indicated that activation of *JNK* under endosomal control can activate downstream transcription factors in the *JNK* pathway, resulting in massive lamellocyte production.” from our paper.

4) Mixture of lymph gland- and circulating hemocyte phenotypesLymph gland- and circulating hemocytes are intermingled, and some of the pathways, including the Toll pathway and autophagy, are validated only by circulating hemocytes. The mixture of two cell types throughout the manuscript makes it hard to follow the authors' rational. Moreover, hemocyte differentiation/proliferation mechanisms in two tissues may differ. The authors need to better align the two populations with more caution and confirm all the pathways in both cell types.

We previously confirmed the Toll pathway in lamellocyte differentiation in lymph glands, showing that Dif played a more important role in Rab5/Rab11^DN^-induced lamellocyte formation (Figure 7I-P). To examine autophagy in lymph glands, we starved *Hml>UAS-GFP* larvae for 4 h, 6 h and 8 h and stained lymph glands with anti-Atg8. Compared with lymph glands from fed larvae, we could not see significant autophagosome formation after starvation (Author response image 2). In contrast, after starvation for 4 h, 6 h or 8 h, we observed obvious autophagosome formation in hemocytes (Author response image 2), showing that hemocytes are more sensitive to starvation and are probably a better model for studying the relationship between autophagy and hematopoiesis. In addition, we also confirmed the *JNK* and Ras/EGFR pathways in lamellocyte formation in circulating hemocytes (Figure 3—figure supplement 2; Figure 6—figure supplement 1).

**Author response image 2. respfig2:** 

5) The non-cell-autonomous phenotype in the lymph glandIn Figure 2H-K, the authors showed that the expression of RNAi against Rab5 or Rab11 in the CZ decreases the MZ in a non-cell-autonomous manner. How do the authors explain this phenotype by the current model? This should be shown with additional experiments.

Based on the current model, CZ-to-MZ progenitor maintenance is mainly regulated by PVR/STAT92E/adenosine signaling (Mondal et al., 2011). To examine whether the non-cell-autonomous regulation from the CZ in our study is dependent on the PVR/STAT92E/adenosine signal, we crossed *UAS-STAT92E* and *UAS-Adgf-A* with *Hml>UAS-Rab5/11^DN^* and stained the lymph gland with anti-P1. However, the increased P1-positive area in *Hml>UAS-Rab5/11^DN^* lymph glands was not restored by *UAS-STAT92E* and *UAS-Adgf-A*, indicating that this non-cell-autonomous regulation from the CZ is independent of the PVR/STAT92E/adenosine signal (Figure 2—figure supplement 2). The underlying mechanism warrants future investigation.

Physiological- or cell biological relevance of Rab5/11-mediated functionWhat is the physiological or cell biological relevance of the Rab5/Rab11 regulation shown in this study? It will be important to show whether the endocytic control indeed is critical for the immune response/hemocyte development to manage all the associated signals.

Rab proteins are important for signal transduction. The inhibition of *Rab5*/*Rab11* leads to aberrant signal transduction and eventually activates multiple innate immunity-associated signaling pathways. Activation of multiple pathways will result in abnormal hematopoiesis, including overproliferation and overdifferentiation of hemocytes. We believe that endocytic control is important for immune response/hemocyte development based on the following:

1) In addition to the *UAS-Rab5 RNAi* and *UAS-Rab11 RNAi* lines, we utilized the *UAS-Rab5^DN^* and *UAS-Rab11^DN^*lines, which are defective for GTPase activity (Zhang et al., 2007). In our study, when we overexpressed *UAS-Rab5/Rab11^DN^* in the blood system by using *Hml-Gal4*, we observed aberrant hematopoiesis, including increased hemocyte counts, an expanded CZ area and massive lamellocyte formation; we confirmed this phenotype using Rab RNAi lines. These results showed that not only the expression level of *Rab5*/*Rab11* but also Rab5/Rab11 GTPase activity play crucial roles in maintaining hematopoietic homeostasis.

2) Many studies have shown that the inhibition of *Rab5/Rab11*, essential genes in endocytosis, can activate innate immunity-related signaling pathways, including the *JNK*, BMP, and Ras/EGFR pathways (Tang et al., 2017; Takino et al., 2014; Bhuin and Roy, 2010). A previous study showed that knocking down *ARF1* or inhibiting ARF1 GTPase resulted in an increased Hrs-positive endosome count and the accumulation of Notch signaling in Hrs-positive endosomes, which eventually activated the Notch pathway (Khadilkar et al., 2014). Similarly, we also observed an increased Hrs-positive endosome count in *Hml>UAS-Rab5/11^DN^*, which means that defective Rab5/Rab11 GTPase led to abnormal endocytic activity. We also showed higher degrees of colocalization between Hrs and p-*JNK* and observed activation of p-*JNK* signaling. These data indicated that the endocytic control of Rab5/Rab11 is crucial for maintaining the homeostasis of signaling pathways.

Tang, Y., Geng, Q., Chen, D., Zhao, S., Liu, X., Wang, Z., 2017. Germline Proliferation Is Regulated by Somatic Endocytic Genes via *JNK* and BMP Signaling in *Drosophila*. Genetics 206(1), 189-197.

Bhuin, T., Roy, J.K., 2010. Rab11 regulates *JNK* and Raf/MAPK-ERK signalling pathways during *Drosophila* wing development. Cell Biol. Int. 34(11), 1113-1118.

Zhang, J., Schulze, K.L., Hiesinger, P.R., Suyama, K., Wang, S., Fish, M., Acar, M., Hoskins, R.A., Bellen, H.J., Scott, M.P., 2007. Thirty-One Flavors of *Drosophila* Rab Proteins. Genetics 176(2), 1307-1322.

[Editors' note: further revisions were suggested prior to acceptance, as described below.]

Reviewer #1:Yu and colleagues supplemented a large number of data and addressed most of the concerns raised by the reviewers. New data clearly resolved concerns raised and explained how circulating hemocytes are increased upon loss of Rab5/11 and why Rab5/11-gal4 differs from Hml-gal4. Despite the clarification, the biological relevance of Rab5/Rab11 signaling in hematopoiesis is still unclear. It is well-known that Rab5/Rab11 endosomal pathway controls multiple proteins involved in cellular signalings, including EGFR/Ras, Eiger/JNK, and Toll, and thus, in my opinion, it will be important to understanding why Rab5/Rab11-mediated control is significant in the blood.1) Biological relevanceAlthough the authors extensively investigated the functions of Rab5/Rab11 in the blood cell signaling, it is still ambiguous how the endosomal control is maintained during development or immunity and why it is significant in hematopoiesis. For example, does Rab5/Rab11 control the proliferation of hemocytes in a specific developmental stage by repressing or activating Ras/Erk? Is Rab5/Rab11 inhibited by immune challenges to directly stimulate the lamellocyte differentiation? In which context Rab5/Rab11 gets modified and manipulates target signals similar to phenotypes derived by Rab5DN/Rab11DN or Rab5/Rab11 overexpression?

We thank the reviewer for these points. We believe that there could be an interaction between Rab5/Rab11 and immunity (the lamellocyte formation upon wasp challenge). We will focus on these issues in our future study.

2) Schematic diagramA schematic diagram shown in Figure 10B does not fully represent the genetic pathways identified in the results. For example, there is no evidence showing a direct activation of pERK by Rab5/Rab11. And even though Rasv12 alone could not activate the lamellocyte differentiation, active Ras and Bsk together can induce the lamellocyte. The authors introduced dFOXO as a target of JNK and showed that loss of dFOXO in Hml>Rab5/11DN rescues the lamellocyte phenotype. These points are missing. Or the diagram could be simplified to help understand the major points.

Given that there is no evidence showing loss of *Rab5*/*Rab11* could directly activate p-Erk, we used the dotted line and one question mark in our revised version to display the relationship between Rab5/Rab11 and p-Erk (Figure 10B). In addition, we showed dFOXO as the target of the *JNK* pathway to facilitate lamellocyte formation (Figure 10B). We also simplified the schematic diagram to make it clearer.